# Analog Circuit Topology Design and Sizing with Flow Matching Graph Learning

## Abstract

The soaring demand for electronic devices calls for novel and more efficient analog circuits design. Deep generative models have shown promise in assisting topology, parameter sizing, and layout design process, but existing approaches treat these tasks separately and lack generalizability across diverse problem settings. In this work we introduce a flow matching model for automatic analog circuit design, which achieves high-quality sampling across a variety of topologies and representations. Our model showcases state-of-the-art performance on end-to-end topology design and sizing on the Open Circuit Benchmark (OCB) dataset, and on transistor-level topology generation on the AnalogGenie dataset. Code and models are provided as external supplementary files to this submission.

## 1 Introduction

The automation of analog circuit design stands as an active area of research, driven both by the demand for increasingly efficient architectures to sustain the growth of the electronics industry and by the intrinsic complexity of the task, which is notoriously more challenging than digital circuit design due to its greater diversity of components. Accordingly, the literature presents a wide range of data-driven approaches aimed at automating one or more steps of the analog design workflow, which traditionally includes topology discovery (Lu et al., 2022; 2023; Poddar et al., 2024), parameter sizing (Wang et al., 2018; 2020; Krylov et al., 2023), and layout prediction (Kunal et al., 2019; Xu et al., 2019; Liu et al., 2025a).

Despite significant progress, several hurdles remain. Many methods exhibit limited generalizability, restricting their applicability to a small set of circuit topologies. Others rely on multiple models trained for different subtasks or require substantial computational resources. The absence of widely adopted benchmarks and open models has also often been cited as a limiting factor for faster advancement in the field. This issue has been partly addressed by the recent release of benchmarks and models targeting topology generation and device sizing (Dong et al., 2023; Gao et al., 2025), enabling more systematic comparisons. In terms of model architectures, the long-standing paradigm of representing circuits as graphs (Ren et al., 2020; Wang et al., 2020; Hakhamaneshi et al., 2022; Shahane et al., 2023) now coexists with the recent adoption of Large Language Models (LLM)-based methodologies (Yin et al., 2024; Liu et al., 2024a;b), which harness the exceptional ability of LLMs for sequence modeling to generate circuit design as textual outputs (Chang et al., 2024; Lai et al., 2025). There is however still room for improvement, while the generalizability of these methods beyond pre-defined settings remains an open question.

We argue that graph-based representations of circuits hold untapped potential to address these limitations. In particular, recent progress in generative modeling of graphs using denoising diffusion (Vignac et al., 2023) and flow matching (Eijkelboom et al., 2024; Qin et al., 2025) lets us foresee promising applications for analog circuit design. These models are notorious for their high sample quality (Esser et al., 2024), and can accommodate the conditional generation of multimodal data, opening the door to applications such as circuit completion or parameter sizing within a single architecture. Diffusion models have already proved successful for device sizing (de Azevedo et al., 2025; Eid et al., 2025) and topology discovery (Liu et al., 2025b), but for a limited scope of circuits. So far, only one study has attempted to tackle these tasks jointly (Hou et al., 2024)

In this work, we introduce a multimodal flow matching model, **CircuitFlow**, for end-to-end generation of analog circuit topology and device sizing. Built on a graph transformer backbone, it

shows remarkable sampling quality and allows a very fine control on the denoising process through a modality-dependent time sampling scheme. We evaluate our approach on established benchmarks: first at the behavioral level on the OCB dataset (Dong et al., 2023), and then at the transistor level on the AnalogGenie dataset (Gao et al., 2025) with a separate model for the prediction of port connectivity, achieving in both cases state-of-the-art improvement in the quality of generated circuits. The main contribution of this work is the extension of multimodal flow matching to analog circuit design, yielding a unified framework that jointly addresses discrete topology generation and continuous device sizing across multiple representation levels and circuit complexities. This is notably enabled by sampling the denoising time index *separately for each dimension*, granting unprecedented flexibility for diverse inference-time applications without the need for additional training. The remainder of this paper is structured as follows. Related work is discussed in Section 2. Section 3 introduces the theoretical foundations of this work. The proposed approach is detailed in Section 4, and experimental results are presented in Section 5.

## 2 RELATED WORK

### 2.1 DATA-DRIVEN TOPOLOGY DESIGN AND SIZING OF ANALOG CIRCUITS

**Topology generation**. Data-driven approaches for topology design include Reinforcement Learning (RL) (Fan et al., 2021; Zhao & Zhang, 2022), Bayesian Optimization (BO) in the continuous latent space of a Variational Auto-Encoder (VAE) (Lu et al., 2022; 2023; Dong et al., 2023; Shen et al., 2024), and retrieval from predefined building blocks (Fayazi et al., 2023; Poddar et al., 2024). RL and BO methods often suffer from slow convergence, while retrieval-based strategies depend heavily on the completeness of predefined architectures and typically lack flexibility. Recent works leverage pre-trained LLMs to generate topologies as text output (Chang et al., 2024; Lai et al., 2025), but have not yet scaled beyond a limited set of circuit types and complexity. AnalogGenie (Gao et al., 2025) demonstrates strong scalability to diverse, transistor-level topologies, but its GPT-based backbone requires extensive data augmentation to enforce permutation invariance over input graph nodes.

**Device Sizing**. Parameter sizing, whether at the behavior or transistor level, has been widely explored using RL (Wang et al., 2018; Settaluri et al., 2020; Wang et al., 2020; Cao et al., 2022; Gao et al., 2023; Cao et al., 2024), BO (Lyu et al., 2018), supervised learning (Hakhamaneshi et al., 2022; Krylov et al., 2023) or LLMs (Yin et al., 2024; Liu et al., 2024a;b). Some works aim to address both topology design and sizing (Fayazi et al., 2023; Lu et al., 2023; Liu et al., 2025b), but do so in several stages with separate, dedicated models. An exception is CktGen (Hou et al., 2024), which addresses these tasks jointly using a VAE model. Their approach is however limited to operational amplifiers, and the absence of released models precludes formal comparison.

### 2.2 FLOW MATCHING FOR GRAPH GENERATION

Flow matching models (Lipman et al., 2023; Esser et al., 2024; Tong et al., 2024) have emerged as a sample-efficient alternative to diffusion models (Sohl-Dickstein et al., 2015; Ho et al., 2020). They have been extended to discrete state-space (Campbell et al., 2024; Gat et al., 2025) and proved successful on diverse graph generation tasks (Eijkelboom et al., 2024; Qin et al., 2025). To date, flow matching has not been applied to analog circuit design. A handful of methods have explored the use of denoising diffusion for sizing and topology discovery (de Azevedo et al., 2025; Eid et al., 2025; Liu et al., 2025b), but remain restricted to a narrow range of circuit topologies.

## 3 PRELIMINARIES

### 3.1 CONTINUOUS FLOW MATCHING

The objective of continuous flow matching is to learn an approximation function $u_t^\theta$ of a quantity $u_t$ called a *vector* or *velocity field*, which, given an arbitrary dimension $d$, is a function of $\mathbb{R}^d$ in itself that describes the instantaneous change of a *flow* $x_t$ with respect to a *time dimension*:

$$u_t(x_t) = \frac{dx_t}{dt}, \text{ with } x_t \in \mathbb{R}^d. \tag{1}$$

Given time-varying probability distribution $p_t$, $u_t$ is said to *generate* the probability path $p_t$ if $x_t$ is a random variable that follows $p_t$, where the *prior* $p_0$ is typically a standard Gaussian or uniform distribution, and $p_1$ is the unknown data distribution. This is expressed by the *continuity equation* which links $u_t$ to $p_t$:

$$\frac{dp_t(x_t)}{dt} = -\text{div}(p_t(x_t)u_t(x_t)), \tag{2}$$

where $\text{div}(f(x)) = \sum_i \frac{\partial f(x)_i}{\partial x_i}$ is the divergence operator. Ideally, for a given $p_t$ that satisfies the continuity equation and a trained approximation function $u_t^\theta$, sampling from the data distribution can be achieved by drawing $x_0$ from the prior and solving Equation (1) up to $t = 1$. In practice, one cannot express $p_t$ and $u_t$ directly, but may instead define them as the expectations of a *conditional* path and velocity field over the data distribution $p_1$ (Lipman et al., 2023):

$$p_t(x_t) = \int p_t(x_t|x_1)p_1(x_1)dx_1, \tag{3}$$

$$u_t(x_t) = \int u_t(x_t|x_1)\frac{p_t(x_t|x_1)p_1(x_1)}{p_t(x_t)}dx_1. \tag{4}$$

This definition allows two important results. The first corollary is that if $u_t(x_t|x_1)$ generates $p_t(x_t|x_1)$, then $u_t$ generates $p_t$. Hence it is enough to define a conditional velocity field and probability path that satisfy the continuity equation, which is a much easier task. Second, the same parameter set $\theta$ minimizes the following objectives:

$$\mathcal{L}_{\text{FM}} = \mathbb{E}_{t,x_t \sim p_t}\left[\|u_t^\theta(x_t) - u_t(x_t)\|_2^2\right], \text{ and } \mathcal{L}_{\text{CFM}} = \mathbb{E}_{t,x_1,x_t \sim p_t(x_t|x_1)}\left[\|u_t^\theta(x_t) - u_t(x_t|x_1)\|_2^2\right]. \tag{5}$$

As $\mathcal{L}_{\text{CFM}}$ offers a tractable objective, it is therefore enough to reason in terms of conditional quantities. In summary, if $p_t(x_t|x_1)$ and $u_t(x_t|x_1)$ are chosen adequately such that $u_t(x_t|x_1)$ generates $p_t(x_t|x_1)$, then minimizing Equation (5) (right) amounts to fitting a neural network $u_t^\theta$ which generates $p_t$, i.e., which may then be used to sample from $p_1$. Lipman et al. (2023) propose to write the conditional probability path as a Gaussian:

$$p_t(x_t|x_1) = \mathcal{N}(x_t; \mu_t(x_1), \sigma_t^2(x_1)I), \tag{6}$$

with $\mu_0(x_1) = 0$, $\mu_1(x_1) = x_1$, $\sigma_0(x_1) = 1$, and to write the flow $x_t$ as $x_t = \sigma_t(x_1)x_0 + \mu_t(x_1)$ where $x_0 \sim p_0$. When $\sigma_t \to 0$, and $\mu_t(x_1) = tx_1 + (1 - t)x_0$, one obtains the well-known *rectified flow* (Liu, 2022):

$$x_t = tx_1 + (1 - t)x_0, \tag{7}$$

$$u_t(x_t|x_1) = x_1 - x_0 = \frac{x_1 - x_t}{1 - t}, \tag{8}$$

which can be used in Equation (5) to compute $\mathcal{L}_{\text{CFM}}$. Once trained, the model can be used to draw from $p_1$ starting from a noise sample $x_0 \sim p_0$ and following denoising Euler steps $\Delta t$:

$$x_{t+\Delta t} = x_t + \Delta t\, u_t^\theta(x_t). \tag{9}$$

## 3.2 DISCRETE FLOW MATCHING

One approach (Campbell et al., 2024) to modeling discrete data $x_1 \in [1, \dots, S]^D$, where each dimension of $x_1$ can take $S$ different states, is to consider the whole flow $x_t$ as discrete, and allow state transitions to occur *one dimension at a time*. This translates in the following factorization of $p_{t+\Delta t}(x_{t+\Delta t}|x_t)$:

$$p_{t+\Delta t}(x_{t+\Delta t}|x_t) = \prod_d p_{t+\Delta t}(x_{t+\Delta t}^d|x_t). \tag{10}$$

This allows to define the generative process using a *rate matrix* $R_t \in \mathbb{R}^{S \times S}$ which characterizes state transition over single dimensions and replaces the velocity field $u_t$ from the continuous setting, as can be seen from the denoising process:

$$x_{t+\Delta t}^d \sim \text{Cat}(\delta\{x_t^d, x_{t+\Delta t}^d\} + \Delta t\, R_t(x_t^d, x_{t+\Delta t}^d)). \tag{11}$$

The rate matrix satisfies $R_t(i, j) \geq 0$ if $i \neq j$ and $R_t(i, i) = -\sum_{j \neq i} R_t(i, j)$. As in the continuous case, $R_t$ must satisfy the continuity equation (known as the Kolmogorov equation in the discrete

case, see Gat et al. (2025)) to ensure that it generates $p_t$, and that solving Equation (11) amounts to sampling from the data distribution $p_1$. Again, it is more convenient to define a *conditional rate matrix* $R_t(x_t, x_{t+\Delta t}|x_1)$ that generates the conditional distribution $p_{t|1}$, and that can be used to recover the marginal rate matrix $R_t$ through the following expectation:

$$R_t(x_t, j) = \mathbb{E}_{p_{1|t}(x_1|x_t)} R_t(x_t, j|x_1). \tag{12}$$

This time however the conditional rate matrix can be computed in closed form, and one instead aims to learn the posterior probabilities $p_{1|t}(x_1|x_t)$. In the multivariate case, each dimension is learned separately, such that the training objective writes:

$$\mathcal{L}_{\text{DFM}} = -\mathbb{E}_{t,x_1,x_t} \sum_d \log(p_{1|t}^\theta(x_1^d|x_t)), \tag{13}$$

with $p_{1|t}^\theta$ the approximation function. Noised vector $x_t$ is sampled per dimension by interpolating between $x_1^d$ and a prior, that we take here equal to the product of marginal probability mass functions over all states, simply written $\text{Cat}(m)$, with $m \in [0,1]^S$ (see Appendix A for the computation of $m$):

$$x_t^d \sim \text{Cat}(t\delta\{x_1^d, x_t^d\} + (1-t) \times m)). \tag{14}$$

Inference is done independently from training, which does not require access to the conditional rate matrix. For the latter, Campbell et al. (2024) introduce the following expression:

$$R_t(x_t^d, x_{t+\Delta t}^d = j|x_1^d) = \frac{\text{ReLU}(\partial_t p_{t|1}(x_{t+\Delta t}^d = j|x_1^d) - \partial_t p_{t|1}(x_t^d|x_1^d))}{S.p_{t|1}(x_t^d|x_1^d)} + R^{\text{DB}}(x_{t+\Delta t}^d = j|x_1^d), \tag{15}$$

where the *detailed balance* term $R^{\text{DB}}$ allows an adjustable level of stochasticity (see Appendix A for the derivation of both terms for our choice of prior distribution). The expectation over $x_1^d$ can be derived in closed form to give the final expression of the marginal rate matrix $R_t$ (see Appendix B):

$$R_t(x_t^d, j) = \frac{\left(1 - m_j + m_{x_t^d}\right)}{S(1-t)m_{x_t^d}} p_{1|t}^\theta(x_1^d = j|x_t)$$

$$+ \frac{\text{ReLU}\left(m_{x_t^d} - m_j\right)}{S(1-t)m_{x_t^d}}(1 - p_{1|t}^\theta(x_1^d = j|x_t) - p_{1|t}^\theta(x_1^d = x_t^d|x_t))$$

$$+ \eta\, p_{1|t}^\theta(x_1^d = x_t^d|x_t) + \eta \frac{t + (1-t)m_{x_t^d}}{(1-t)m_j} p_{1|t}^\theta(x_1^d = j|x_t), \tag{16}$$

where $\eta \in \mathbb{R}^+$ is the tunable noise level.

**Multimodal Flows.** In the case of a multimodal flow $(x_t, y_t)$, Campbell et al. (2024) showed that if the noising process $p_{t|1}(x_t, y_t|x_1, y_1)$ factorizes over its variables such that:

$$p_{t|1}(x_t, y_t|x_1, y_1) = \prod_d^{D_x} p_{t|1}(x_t^d|x_1^d) \prod_d^{D_y} p_{t|1}(y_t^d|y_1^d), \tag{17}$$

then the process composed separately of $R_t^x$ (respectively $u_t^x$ if $x_t$ is continuous) and $R_t^y$ (resp. $u_t^y$), as defined above, generates the marginal multimodal flow $p_t(x_t, y_t)$. This allows to sample $t$ *independently for each variable*, which enables a remarkable flexibility of the denoising process.

## 4 GRAPH FLOW MATCHING FOR ANALOG TOPOLOGY DESIGN AND SIZING

For the remainder of this work, we represent circuits as undirected graphs $\mathcal{G}$ composed of a set of $D_v$ nodes $\mathcal{V}$, a set of edges $\mathcal{E} \subseteq \mathcal{V} \times \mathcal{V}$ that connect the nodes, and when applicable, a node feature vector $\mathcal{F}$ that provides component sizes. Individual node elements and node features are respectively noted $v^d$ and $f^d$, $\forall d \leq D_v$, and take values in $\{1, \ldots, S\}$ and $\mathbb{R}$, respectively, where $S$ is the total number of node types. Individual edges are noted $e^d$, $\forall d \leq D_e = D_v \times D_v$, and take values in $\{0, 1\}$. The objective of this work is to train a multimodal flow matching model to sample from the data distribution $p_1(\mathcal{G})$. This section first defines this flow along with its training process. We then describe the architecture of the employed graph transformer model, and give details about the chosen data representation.

## 4.1 MULTIMODAL FLOW FOR FLEXIBLE CIRCUIT MODELING

The flow we consider here is composed of two discrete variables, node types $\mathcal{V}_t$ and edges $\mathcal{E}_t$, along with the continuous device sizes $\mathcal{F}_t$. As before, the noising process will factorize over variables, and as suggested by Campbell et al. (2024) the noising time index will be sampled independently for each variable, yielding respectively $t_v$, $t_e$ and $t_f$ for node types, edges and features. Here however we go one step further and sample time *independently for each dimension*, such that $t_v, t_f \in [0, 1]^{D_v}$ and $t_e \in [0, 1]^{D_e/2}$ (graphs are undirected and only half of the edges are modeled). As we shall see shortly, this makes the sampling process particularly flexible, allowing a whole range of key applications. The noising process thus writes, using the notations $t = (t_v, t_e, t_f)$ and $\mathcal{G}_t = (\mathcal{V}_{t_v}, \mathcal{E}_{t_e}, \mathcal{F}_{t_f})$:

$$p_{t|1}(\mathcal{G}_t|\mathcal{G}_1) = \prod_d^{D_v} p_{t_v|1}(v_{t_v}^d|v_1^d) p_{t_f|1}(f_{t_f}^d|f_1^d) \prod_d^{D_e/2} p_{t_e|1}(e_{t_e}^d|e_1^d), \tag{18}$$

where we omitted the dimension dependency on time for the sake of clarity. Each dimension is noised independently according to Equations (14) and (7) for discrete and continuous variables, respectively, and this can be seen as an extension of the previous multimodal flow where every dimension is a variable on its own. From Proposition 4.2 of Campbell et al. (2024), we know that the following process generates $p_t(\mathcal{G}_t) = \mathbb{E}_{p_1(\mathcal{G}_1)}[p_{t|1}(\mathcal{G}_t|\mathcal{G}_1)]$:

$$R_t(v_{t_v}^d, j) = \mathbb{E}_{p_{1|t}^\theta(v_1^d|\mathcal{G}_t)}[R_t(v_{t_v}^d, j|v_1^d)], \tag{19}$$

$$R_t(e_{t_e}^d, j) = \mathbb{E}_{p_{1|t}^\theta(e_1^d|\mathcal{G}_t)}[R_t(e_{t_e}^d, j|e_1^d)], \tag{20}$$

$$u_t(f_{t_f}^d) = \mathbb{E}_{p_{1|t}^\theta(f_1^d|\mathcal{G}_t)}[u_t(f_{t_f}^d|f_1^d)], \tag{21}$$

where we abused notations and simply referred to all marginal and conditional rate matrices as $R_t$ and $R_t(.|\mathcal{G}_1)$ to keep notations uncluttered. The denoising process for $v_{t_v}^d$ and $e_{t_e}^d$ is done according to Equation (11), where the two rate matrices $R_t(v_{t_v}^d, j)$ and $R_t(e_{t_e}^d, j)$ are computed in closed form using Equation (16) and their respective marginal prior distributions. The denoising process for features $f_{t_f}^d$ follows Equation (9), using the vector field $u_t^\theta(f_t^d)$ learned by the model.

**Loss function.** Our generative model is trained by minimizing $\mathcal{L}_{\text{CFM}}$ on continuous variables and $\mathcal{L}_{\text{DFM}}$ on discrete variables. The overall loss function writes:

$$\mathcal{L} = \mathbb{E}_{t,\mathcal{G}_1,\mathcal{G}_t}\Big[-\sum_d^{D_v} \log(p_{1|t}^\theta(v_1^d|\mathcal{G}_t)) - \sum_d^{D_e/2} \log(p_{1|t}^\theta(e_1^d|\mathcal{G}_t)) + \sum_d^{D_v} \|u_t^{\theta,d}(\mathcal{G}_t) - u_t(f_{t_f}^d|f_1^d)\|_2^2\Big]. \tag{22}$$

---

**Algorithm 1** Training

**Input:** Graph dataset $\mathcal{D} = \{\mathcal{G}^1, \ldots, \mathcal{G}^M\}$
**for** $e = 1$ **to** Max training epoch $E$ **do**
    Sample $\mathcal{G}_1 \sim \mathcal{D}$, $t \sim \mathcal{U}[0, 1]^{2D_v+D_e}$
    Sample $\mathcal{G}_t \sim p_{t|1}(\mathcal{G}_t|\mathcal{G}_1)$     ▷ *Noising*
    $p_{1|t}^\theta(v_1, e_1|\mathcal{G}_t), u_t^\theta(\mathcal{G}_t) \leftarrow f_\theta(\mathcal{G}_t, t)$     ▷
*Forward pass*
    $\mathcal{L}_v \leftarrow \log p_{1|t}^\theta(v_1|\mathcal{G}_t)$     ▷ *Node loss*
    $\mathcal{L}_e \leftarrow \log p_{1|t}^\theta(e_1|\mathcal{G}_t)$     ▷ *Edge loss*
    $\mathcal{L}_f \leftarrow \|u_t^\theta(\mathcal{G}_t) - \frac{f_1-f_t}{1-t}\|^2$ ▷ *Features loss*
    Update $f_\theta$ weights
**end for**

**Algorithm 2** Inference

**Input:** Number of nodes $D_v$
Sample $\mathcal{G}_0 \sim p_0(\mathcal{G}_0)$     ▷ *Sample from prior*
**for** denoising step $t = 0$ to 1 with step $\Delta t$ **do**
    $p_{1|t}^\theta(\mathcal{G}_1|\mathcal{G}_t) \leftarrow f_\theta(\mathcal{G}_t, t)$     ▷ *Denoising*
    $R_t(v_t, .), R_t(e_t, .) \leftarrow$ 19, 20     ▷
*Marginalize rate matrices*
    $v_{t+\Delta t} \sim \text{Cat}(\delta\{v_t, j\} + \Delta_t R_t(v_t, j))$
    $e_{t+\Delta t} \sim \text{Cat}(\delta\{e_t, j\} + \Delta_t R_t(e_t, j))$
    $f_{t+\Delta t} \leftarrow f_t + \Delta t.u_t^\theta(f_t)$     ▷ *Update $\mathcal{G}_t$*
**end for**
**Return:** $\mathcal{G}_1$

---

**Flexible denoising.** The main advantage of the proposed framework is that it allows all dimensions, i.e., components or groups of components, to be denoised independently. Very diverse applications are therefore possible with the same model depending on the chosen *time sampling scheme*:

Circuit completion

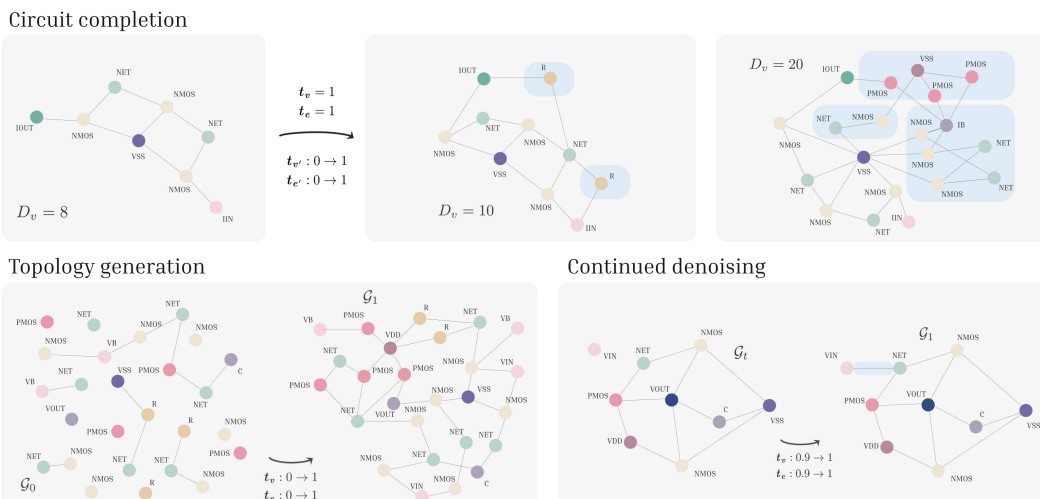

Topology generation

Continued denoising

Figure 1: Multiple applications enabled by our framework, here on the AnalogGenie dataset.

(a) *End-to-end topology design and sizing* $- t_v, t_e, t_f : 0 \rightarrow 1$;

(b) *Circuit completion / conditional inpainting* $- \mathcal{G} = [(\mathcal{V}, \mathcal{E}, \mathcal{F}), (\mathcal{V}', \mathcal{E}', \mathcal{F}')]$ such that $t_v, t_e, t_f = 1$, and $t_{v'}, t_{e'}, t_{f'} : 0 \rightarrow 1$;

(c) *Topology-conditional sizing* $- t_v, t_e = 1$, and $t_f : 0 \rightarrow 1$;

(d) *Link prediction* $- t_v = 1$, and $t_e : 0 \rightarrow 1$.

Another noteworthy application is *continued denoising*, where an output graph $\mathcal{G}$ undergoes additional denoising steps if it fails to meet predefined criteria. By fixing $t$ between 0 and 1 independently for each device and edge, one can control the extent to which one part of the circuit must be denoised further or preserved, allowing a very fine-grained supervision. Illustrations of several applications can be found in Figure 1 and Appendix F, while the overall training and sampling procedures are summarized in Algorithm 1 and 2, respectively.

## 4.2 NETWORK ARCHITECTURE

We base our denoising network on the graph transformer architecture of Ma et al. (2023) which combines a global receptive field with an expressive random walk structural encoding, achieving strong performance across diverse graph learning tasks. To adapt it to our framework, we introduce two key modifications. First, we incorporate a time-conditioning mechanism, inspired by Diffusion Transformers (Peebles & Xie, 2023), mapping $t_v$ (and respectively $t_e$ and $t_f$) into multiplicative and additive biases $\alpha_v$, $\alpha'_v$, $\beta_v$, $\beta'_v$, $\gamma_v$, and $\gamma'_v$, applied at different stages of each transformer layer. Second, we add a dedicated processing path for node features $f$, mirroring the sequence of operations used for node types $v$. The graph transformer layer of CircuitFlow is illustrated in Figure 2. Overall, our model is a light architecture of 2.05M parameters.

## 4.3 CONDITIONAL GENERATION

In practice, analog circuit design typically supposes to adhere to predefined performance objectives. We present here how such control can be achieved, through two different guidance approaches (Nisonoff et al., 2025). Classifier guidance (CG) requires training a classifier model $p^\phi(c|\mathcal{G}_t, t)$ to predict the class of the conditioning signal $c$, given a noised

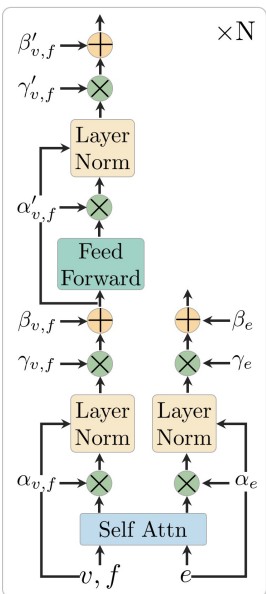

Figure 2: The conditional graph transformer layer of CircuitFlow.

Figure 3: Limitations of the OCB/CktGNN representation. Without explicit nets, graphs are more complex **(left)**, and ambiguous **(right)**: the output of the second single-stage op-amp (++), on the left side of the right panel, is *not* linked to the input of the third op-amp (+−), which belongs to a *feedback path*: arrows direction should be reversed. The DAG representation wrongly portrays the circuit as valid: it is *open* and therefore invalid, as shown when the feedback path edges are reversed.

sample $\mathcal{G}_t$. It is then used during the denoising process to bias the rate matrix of an *unconditional* generative model, yielding a *guided* rate matrix $R_t^{(\gamma)}(.|c)$, where $\gamma$ controls the *guidance strength*:

$$(CG): \quad R_t^{(\gamma)}(x_t, j|c) = \left[ \frac{\log p^\phi(c|j,t)}{\log p^\phi(c|x_t,t)} \right]^\gamma R_t(x_t, j). \tag{23}$$

This necessitates evaluating $p^\phi$ over all state transitions, which can become costly. An alternative is to use classifier-free guidance (CFG), which only requires two passes through the model at a denoising step $t$, one with the conditioning signal $c$ yielding the conditional rate matrix $R_t(.|c)$, and one where $c$ is replaced by a mask token $\emptyset$, yielding the unconditional $R_t$. The mask token is learned during training by randomly masking $c$ with a chosen probability. In CFG, $R_t^{(\gamma)}(.|c)$ writes:

$$(CFG): \quad R_t^{(\gamma)}(x_t, j|c) = R_t(x_t, j|c)^\gamma R_t(x_t, j)^{1-\gamma}. \tag{24}$$

The previous procedure is applied to node types and edges. For the continuous case of node features, the unconditional velocity field $u_t^\theta(f_t|f_1)$ is updated at each step $t$ using one of the following expressions (see Dhariwal & Nichol (2021) and Ho & Salimans (2022)):

$$(CG): u_t^{(\gamma)}(f_t|f_1, c) = u_t^\theta(f_t|f_1) + \gamma \nabla_{f_t} \log p^\phi(c|\mathcal{G}_t, t), \text{ or}$$

$$(CFG): u_t^{(\gamma)}(f_t|f_1, c) = u_t^\theta(f_t|f_1, \emptyset) + \gamma.(u_t^\theta(f_t|f_1, c) - u_t^\theta(f_t|f_1, \emptyset)).$$

### 4.4 UNIFIED CIRCUIT GRAPH REPRESENTATION

Topology design and device sizing can be performed at different levels of the circuit representation hierarchy, including the behavior and the transistor level. In the former case, transistors do not appear explicitly, but are included in larger substructures, such as single-stage operational amplifiers (op-amps). Following prior work (Ren et al., 2020; Hakhamaneshi et al., 2022) we represent circuit and voltage nodes (i.e., nets and ports) as graph nodes, along with circuit devices. This results in a unified circuit representation that accommodates both representation levels, while conforming to the writing conventions of the SPICE simulation software. At the transistor level, we do not represent transistor pins explicitly, but proceed in a *hierarchical approach* to predict their connectivity. As noted by Gao et al. (2025), this is necessary to disambiguate topologies which do not specify pin assignments. To achieve this, we train a dedicated model to regress edge probabilities between transistors and their neighboring nodes, based on the output of a topology generation model. This approach ensures a one-to-one mapping between generated graphs and SPICE netlists across all representation levels. Further details on the hierarchical model are provided in Appendix C.

## 5 EXPERIMENTS

We evaluate the performance of CircuitFlow on two standard benchmarks for circuit topology generation: OCB (Dong et al., 2023) and AnalogGenie (Gao et al., 2025), which represent circuits at distinct abstraction levels. Since OCB also provides device sizes, we additionally assess the model's effectiveness on the sizing task on this dataset. Both datasets however require preprocessing to map circuits into our unified representation, which we describe in the following section. All code, model weights and processed datasets are made publicly available.

| Model | V.U.N.↑ | Val. sim.↑ | Val. graphs↑ | Val. circuits↑ | Uniqueness ↑ | Novelty ↑ |
|---|---|---|---|---|---|---|
| DAGNN | – | – | 83.1 | 74.2 | – | **97.2** |
| PACE | – | – | 83.1 | 75.1 | – | 97.1 |
| D-VAE | 44.5 | 58.1 | 67.7 | 59.5 | 84.3 | 94.5 |
| CktGNN | 47.9 | 74.2 | 85.1 | 81.4 | 72.6 | 93.0 |
| **CircuitFlow** (Ours) | **74.3** | **92.9** | **99.4** | **98.4** | **85.8** | 91.1 |

Table 1: Evaluation of output topology quality across various architectures on OCB. Results that could not be reproduced are reported from Dong et al. (2023). All metrics are expressed in **percentage**, and uniqueness is computed over $10,000$ samples.

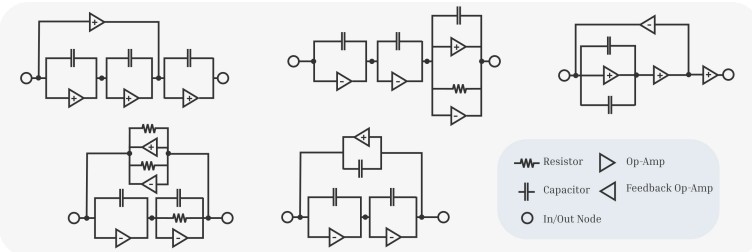

Figure 4: Novel op-amp circuit topologies produced by CircuitFlow. Certain samples possess components (parallel capacitors or resistors) that make them equivalent to a circuit with fewer nodes.

## 5.1 DATASETS AND PREPROCESSING

**OCB.** The OCB dataset consists of 10,000 DAGs describing up to 3-stage op-amps at the *behavioral level*, split into 9,000 training samples (composed of 3,957, or $44\%$, unique topologies) and 1,000 test samples. Each graph comes in two versions, one where components are grouped using a predefined set of *subgraphs*, and one decomposed into individual components, the latter including device sizes. To keep the approach general we work at the component level, though several preprocessing steps are more conveniently done at the subgraph level, which requires to map the circuit back to its component and recover the sizes.

The first of these steps is to *identify and revert feedback path edges*: as illustrated in Figure 3 (right), using DAGs leads to *ambiguous topologies*, preventing the identification of feedback path components and misrepresenting their input/output pin assignment. We therefore revert all input and output edges to subgraphs containing feedback op-amps ($gm$- in OCB terminology). Following common practice, we then add *circuit nets* as graph nodes, fully disambiguating device connections. This also simplifies the graphs (Figure 3, left). Finally DAGs are converted into undirected graphs, and we validate the robustness of the whole process by ensuring that all circuits remain simulatable.

**AnalogGenie.** The AnalogGenie dataset contains 3,350 samples spanning 11 analog circuit types (op-amps, SC-samplers, bandgap references, power converters, etc.). Circuits are represented at the transistor level and are substantially larger than OCB samples (see Table 7 in Appendix D for an exhaustive comparison). Gao et al. (2025) highlight the need to represent transistor pins to disambiguate otherwise similar topologies, but omit net nodes. This has two detrimental consequences: (i) as discussed in the previous section, it creates unnecessary complexity: adding net nodes reduces mean graph density from 0.09 to 0.04; (ii) it breaks the invariance between the pins of symmetric devices (resistors, capacitors, inductors), which must therefore be learned. We therefore preprocess AnalogGenie circuits by adding net nodes, and removing pin nodes for symmetric devices.

## 5.2 BEHAVIOR-LEVEL TOPOLOGY GENERATION AND SIZING ON THE OCB DATASET

**Topology Generation.** We first evaluate the quality of circuit topologies generated by CircuitFlow, using several graph learning baselines: D-VAE (Zhang et al., 2019), DAGNN (Thost & Chen, 2021), PACE (Dong et al., 2022), and CktGNN (Dong et al., 2023). D-VAE and CktGNN were retrained using the code provided by Dong et al. (2023). We apply the same processing to CktGNN's outputs as described above for OCB subgraphs to reflect actual circuit topologies. Metrics from

| Accuracy (%) ↑ | Gain | UG-$f$ | PM | Joint |
|---|---|---|---|---|
| **CFG** | | | | |
| *Topology* | 50.3 | 24.3* | 32.2 | 6.83 |
| *Topo. + sizing* | 62.3 | 28.0* | 33.2 | 7.28 |
| **CG** | | | | |
| *Topo. + sizing* | **65.7** | **61.0** | **73.3** | **28.9** |

Table 2: Conditional generation accuracies on gain, unit-gain frequency (UG-$f$) and phase margin (PM), along with joint accuracy for two conditioning methods, classifier-free guidance (**CFG**) and classifier guidance (**CG**). Results marked with $*$ are undistinguishable from random sampling under a two-sided binomial test ($p$-value $> 0.05$).

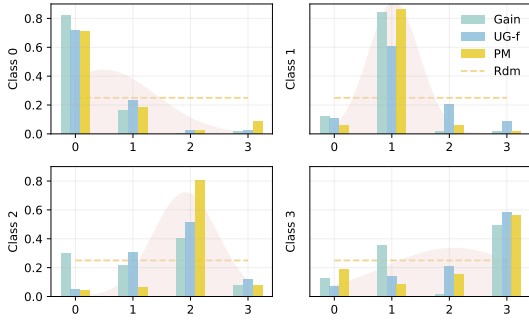

Figure 5: Distribution of output circuit features for the four conditioning categories and all three specifications, using CG. This shows the model has learned to conform to the required spec values.

this work are used here, including **Valid graphs** (percentage of connected graphs with input and output nodes) and **Valid circuits** (circuit with a main path composed only of single-stage op-amps). However, these do not capture graphs with nodes of degree one, which indicates open circuits, and ignore feedback paths. This means that some circuits that appear valid are not simulatable. We therefore add a **Valid sim** metric, similarly to Gao et al. (2025), that gives the proportion of circuits simulatable with SPICE with default parameters. Finally we also report the **V.U.N** (Vignac et al., 2023), which gives the fraction of outputs that are simultaneously simulatable, unique, and novel. Results from Table 1 show the exceptional ability of CircuitFlow to produce novel and valid outputs using as few as 100 denoising Euler steps, improving on CktGNN by 18 points in Valid sim and 26 points in V.U.N, hence establishing a new state-of-the-art on the OCB dataset. Examples of novel generated topologies can be found in Figure 4.

**Conditional Circuit Design.** We now examine how guidance can be used to control both topology design and device sizing based on predefined performance specifications. The conditioning signal $c$ is a triplet $(c_g, c_{pm}, c_{ugf})$ corresponding to conditioning gain, phase margin and unit-gain frequency, following the features from OCB. Each quantity is discretized into quartiles so that the resulting four bins are equally likely. Marginal and joint accuracies between output and conditioning quantities are reported in Table 2, for both topology generation alone and full circuit design. Statistical significance is evaluated with two-sided binomial tests. For CFG we set $\gamma = 2$ for all modalities, and use $\gamma = 15$ for nodes and 30 for features for CG, and omit conditioning on edges. Results on topology generation provide an important insight: circuit topology is correlated with circuit-level specifications, even with randomized features. Thus topology discovery must be performed *conditionally* on circuit specifications, in contrast to previous approaches (Dong et al., 2023; Gao et al., 2025). Accuracy is further improved when jointly learning topology and sizes, and full conditional design with CG achieves *28.9% joint accuracy* over all 64 test categories. We represent in Figure 5 the marginal distributions of output classes for all four conditioning categories of each specification, showing the effectiveness of our conditioning method. Conditional generation examples can be found in Figure 8 in appendix.

### 5.3 TRANSISTOR-LEVEL TOPOLOGY GENERATION ON THE ANALOGGENIE DATASET

This section explores the scalability of CircuitFlow to the more complex and diverse graphs of the AnalogGenie benchmark. Following Gao et al. (2025), we compare with Lamagic Chang et al. (2024) and AnalogCoder (Lai et al., 2025) which, while both trained on distinct datasets and being limited in terms of circuit complexity, represent important milestone works. We train CircuitFlow with our hierarchical approach, using a dedicated model for the regression of pin assignment probabilities. The number of denoising Euler steps is kept to 100. This time we explore using continued denoising as a post-processing strategy. To this end we select invalid circuits based on predefined but simple rules: disconnected graphs, absence of a *VSS* node, or node degree inconsistent with device pin number. Those circuits then undergo 5 additional denoising steps starting from $t = 0.9$,

ensuring that most of the topology is preserved (examples of this process can be found in Figure 11 in appendix). The whole process, which can be seen as an adaptive denoising stop condition, is repeated up to 5 times, obtaining significant improvements at a negligible cost. Results can be found in Table 3. Overall, we achieve a new state-of-the-art on the AnalogGenie dataset over all considered metrics, improving V.U.N over AnalogGenie by 13 to 26 points. Notably, this is done with minimal inductive bias, and a very light preprocessing pipeline. Finally, we report the Jensen-Shannon divergence between output and data distributions of node types ($JS_{data}$), showing that our method has learned to match the node type distribution much more closely, hinting at a more faithful coverage of the diversity of the training dataset. Examples of output circuits can be found in Figures 9 and 10.

| Model (all metrics ↑) | V.U.N. (%) | Val. sim (%) | Uniqueness (%) | Novelty (%) | Max Node Nb | $JS_{data}$ ($\times 10^{-3}$) ↓ |
|---|---|---|---|---|---|---|
| LaMAGIC | – | 68.2 | – | 12.7 | 4 | - |
| AnalogCoder | – | 57.3 | – | 8.9 | 10 | - |
| AnalogGenie | 62.2 | 73.1 | 88.5 | **100** | 63 | 14.5 |
| **CircuitFlow** (Ours) | 75.6 | 75.7 | **98.9** | **100** | **71** | 2.6 |
| **CircuitFlow** + post-pro. | **88.0** | **88.1** | 98.7 | **100** | **71** | **1.8** |

Table 3: Output topology quality on the AnalogGenie dataset. The Jensen-Shannon divergence between data and output node type distributions measures the diversity of generated architectures.

## 5.4 ABLATION STUDIES

We evaluate the effect of time sampling granularity by applying graph-, modality- and dimension-wise time sampling both at training and inference (Table 4). Our findings indicate that sampling time per modality or dimension during training leads to higher validity. The latter also improves the validity on the circuit completion task, where *sampling time per dimension at inference is necessary*: here, the denoising time index of conditioning nodes and edges needs to remain fixed to one, while only the time index of new nodes can vary. Next, we study how the number of denoising Euler steps affects topology quality on the AnalogGenie dataset (Table 5). Using as few as 50 denoising steps yields results that surpass previous state-of-the art, while 100 steps increase the V.U.N. by 3 additional points. Adding more denoising steps beyond that point does not improve results further.

| $t$ granularity | **Inference** | | |
|---|---|---|---|
| | Topology Gen. | | Completion |
| **Train** | Graph | Mod. | Dim. |
| Graph | 80.5 | 81.0 | 56.5 |
| Modality | 92.6 | 92.7 | 65.7 |
| Dimension | 92.9 | 92.7 | 68.7 |

Table 4: Valid sim (%) per $t$ sampling granularity (graph, modality or dimension level) on unconditional topology generation and circuit completion on the OCB dataset. Circuit completion requires dimension-wise time sampling.

| Euler steps | V.U.N. (%) | Latency ($s$) |
|---|---|---|
| 20 | 58.1 | 0.20 |
| 50 | 72.8 | 0.47 |
| 100 | 75.6 | 0.96 |
| 200 | 74.9 | 1.95 |
| AnalogGenie | 62.2 | 13.1 |

Table 5: Influence of the number of Euler steps on V.U.N. and latency (per sample) on the AnalogGenie dataset.

## 6 CONCLUSION

This work introduces CircuitFlow, a flow matching framework for joint analog circuit topology generation and device sizing. By leveraging independent time sampling across dimensions and modalities, the model achieves remarkable inference-time flexibility, enabling applications such as circuit completion, error correction or link prediction. Experiments show that CircuitFlow consistently produces valid, novel, and simulatable circuits, outperforming prior state-of-the-art models across both considered benchmarks. These results demonstrate its scalability from behavioral-level op-amps to diverse transistor-level circuits, while requiring only minimal preprocessing and inductive bias. Importantly, we also show that joint conditional topology generation and sizing is necessary to achieve a fine-grained control over the performances of the output circuits. This work therefore opens new research directions in generative circuit design, achieving together tasks that were previously treated in isolation.

REPRODUCIBILITY STATEMENT

All results reported here are fully reproducible using the provided code and weights, and the precise pipeline will be described on the project's github page. Likewise, we also realease the preprocessing code for both dataset and for the outputs from the CktGNN model, together with the exact dataset versions used to train our models.

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

## A    DERIVATION OF THE CONDITIONAL RATE MATRIX EXPRESSION

In this paper, we consider the prior distribution over node types and edges as the joint marginal distribution over all $S$ states, where $S$ is the number of node types for nodes and 2 for edges. In the following, we write indiscriminately the variable of interest as $x_t \in [1, \ldots, S]^D$.

We have then:

$$x_0 \sim \text{Cat}(m_1, m_2, \ldots, m_S), \tag{25}$$

where:

$$\forall i, \, m_i \in [0, 1] \quad \text{and} \quad m_i = \frac{1}{ND} \sum_{x \in X_{\text{data}}} \sum_d \delta\{x^d, i\} \tag{26}$$

Now the noised vector $x_t^d$ is sampled from the distribution obtained by interpolating between the one-hot distribution in $x_1^d$ and the prior $p_0$.

$$x_t^d \sim \text{Cat}(t\delta\{x_1^d, x_t^d\} + (1 - t) \times m)) \tag{27}$$

The derivation of the conditional rate matrix is inspired from the uniform case of Campbell et al. (2024), substituting the uniform distribution with the marginal one. We first need to derive $\partial_t p_{t|1}(x_t^d | x_1^d)$ in order to compute the first term of the conditional rate matrix in Equation (15).

$$\partial_t p_{t|1}(x_t^d | x_1^d) = \partial_t \left( t\delta\{x_t^d, x_1^d\} + (1 - t)m_{x_t^d} \right) \tag{28}$$

$$= \delta\{x_t^d, x_1^d\} - m_{x_t^d} \tag{29}$$

Thus, for $x_t^d \neq j$ (diagonal entries will be computed later):

$$R_t^*(x_t^d, j \mid x_1^d) = \frac{\text{ReLU}\left(\partial_t p_{t|1}(j \mid x_1^d) - \partial_t p_{t|1}(x_t^d \mid x_1^d)\right)}{S p_{t|1}(x_t^d \mid x_1^d)} \tag{30}$$

$$= \frac{\text{ReLU}\left(\delta\{j, x_1^d\} - m_j - \delta\{x_t^d, x_1^d\} + m_{x_t^d}\right)}{S\left(t\,\delta\left\{x_t^d, x_1^d\right\} + (1 - t)m_{x_t^d}\right)}. \tag{31}$$

This simplifies as:

$$R_t^*(x_t^d, j \mid x_1^d) = \frac{\left(1 - m_j + m_{x_t^d}\right)}{S(1 - t)m_{x_t^d}}\delta\{j^d, x_1^d\}(1 - \delta\{x_t^d, x_1^d\})$$

$$+ \frac{\text{ReLU}(m_{x_t^d} - m_j)}{S(1 - t)m_{x_t^d}}(1 - \delta\{j, x_1^d\})(1 - \delta\{x_t^d, x_1^d\}). \tag{32}$$

We turn now to the derivation of the *detailed balance* term of the conditional rate matrix, which allows to inject stochasticity in the denoising process. As per Campbell et al. (2024), to ensure that the rate matrix still obeys the continuity equation, $R_t^{\text{DB}}$ must satisfy the following detailed balance condition:

$$p_{t|1}(i|x_1^d)R_t^{\text{DB}}(i, j|x_1^d) = p_{t|1}(j|x_1^d)R_t^{\text{DB}}(j, i|x_1^d) \tag{33}$$

Following their general recipe for DB rate matrix expression and again considering $i \neq j$, we assume:

$$R_t^{\text{DB}}(i, j|x_1^d) = a_t \delta\{i, x_1^d\} + b_t \delta\{j, x_1^d\} \tag{34}$$

Substituting this into Equation (33) yields:

$$\left(t\delta\{i, x_1^d\} + (1-t)m_i\right)\left(a_t\delta\{i, x_1^d\} + b_t\delta\{j, x_1^d\}\right) \tag{35}$$

$$= \left(t\delta\{j, x_1^d\} + (1-t)m_j\right)\left(a_t\delta\{j, x_1^d\} + b_t\delta\{i, x_1^d\}\right). \tag{36}$$

As this must be true for all $i$ and $j$ as long as $i \neq j$, one may fix $i = x_1^d$ to force a simpler relation between $a_t$ and $b_t$:

$$b_t = a_t \frac{t + (1-t)m_i}{(1-t)m_j} \tag{37}$$

In the following $a_t$ is set to a *noise level* $\eta$ which can be seen as a re-noising rate of clean data. Finally, the detailed balance conditional rate matrix writes:

$$R_t^{\mathrm{DB}}(i, j | x_1^d) = \eta\delta\{i, x_1^d\} + \eta\frac{t + (1-t)m_i}{(1-t)m_j}\delta\{j, x_1^d\}. \tag{38}$$

## B  MARGINALIZATION OF THE CONDITIONAL RATE MATRIX

The unconditional rate matrix $R_t(x_t^d, j)$ can be computed in closed form by marginalizing the overall conditional rate matrix $R_t(x_t^d, j \mid x_1^d) = R_t^*(x_t^d, j \mid x_1^d) + R_t^{\mathrm{DB}}(x_t^d, j \mid x_1^d)$ over $x_1^d$, where the probabilities $p_{1|t}^\theta(x_1^d \mid x_t)$ are output by the learned flow matching model $\theta$:

$$R_t(x_t^d, j) = \mathbb{E}_{p_{1|t}^\theta(x_1^d | x_t)}\left[R_t^*(x_t^d, j \mid x_1^d) + R_t^{\mathrm{DB}}(x_t^d, j \mid x_1^d)\right] \tag{39}$$

$$= \mathbb{E}_{p_{1|t}^\theta(x_1^d | x_t)}\left[\frac{\left(1 - m_j + m_{x_t^d}\right)}{S(1-t)m_{x_t^d}}\delta\{j, x_1^d\}(1 - \delta\{x_t^d, x_1^d\}) \right.$$
$$+ \frac{\mathrm{ReLU}(m_{x_t^d} - m_j)}{S(1-t)m_{x_t^d}}(1 - \delta\{j, x_1^d\})(1 - \delta\{x_t^d, x_1^d\})$$
$$\left. + \eta\delta\{x_t^d, x_1^d\} + \eta\frac{t + (1-t)m_{x_t^d}}{(1-t)m_j}\delta\{j, x_1^d\}\right]. \tag{40}$$

Integrating over $x_1^d$ yields the final expression for the marginal rate matrix, wherein each term can be easily computed:

$$R_t(x_t^d, j) = \frac{\left(1 - m_j + m_{x_t^d}\right)}{S(1-t)m_{x_t^d}}p_{1|t}^\theta(x_1^d = j | x_t)$$
$$+ \frac{\mathrm{ReLU}\left(m_{x_t^d} - m_j\right)}{S(1-t)m_{x_t^d}}(1 - p_{1|t}^\theta(x_1^d = j | x_t) - p_{1|t}^\theta(x_1^d = x_t^d | x_t))$$
$$+ \eta\, p_{1|t}^\theta(x_1^d = x_t^d | x_t) + \eta\frac{t + (1-t)m_{x_t^d}}{(1-t)m_j}p_{1|t}^\theta(x_1^d = j | x_t) \tag{41}$$

Diagonal entries of $R_t$ are then obtained following $R_t(i, i) = -\sum_{j \neq i} R_t(i, j)$.

We can finally compute the transition probabilities

$$p_{t+\Delta t | t}(x_{t+\Delta t}^d = j \mid x_t) = \delta\{j, x_t^d\} + R_t(x_t^d, j)\Delta t, \tag{42}$$

which, when done on all dimensions $d$ and all time steps $t$, finally allows to sample new data points from $p_1$.

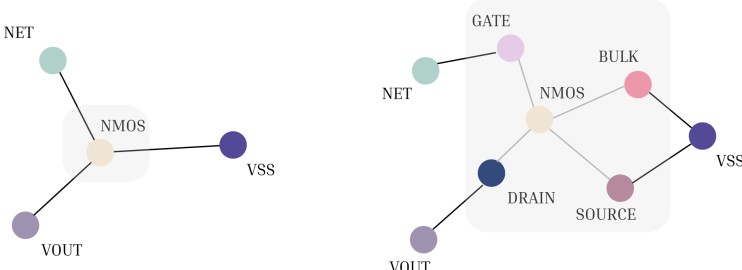

Figure 6: (Left) Device-level circuit, where the NMOS is treated as a single node connected to three nets (NET, VOUT, and VSS). (Right) The corresponding pin-level graph, with explicit pin connections.

## C   PIN-LEVEL PREDICTION

Our two-stage approach for pin-level topology generation involves a first model which generates a circuit topology with device interconnections. Based on the output of the first stage, a second model determines how pins connect to the neighboring components. The process is represented in Figure 6.

The pin assignment model shares the same architecture as the topology generation model, but is trained to regress edge probabilities between a node's pins and the neighboring net nodes. This is done using a classical binary cross entropy objective:

$$\mathcal{L}_{\text{PIN}} = -\frac{1}{|\mathcal{E}_L|} \sum_{e \in \mathcal{E}_L} y_e \log(\hat{y}_e) + (1 - y_e) \log(1 - \hat{y}_e), \tag{43}$$

where $y_e = 1$ if edge $e$ exists, and $0$ otherwise. At inference, assigning transistor pins to their neighbors requires to solve an assignment problem, using the (opposite of the) predicted edge probabilities as the cost function, and fulfilling the constraint that all neighboring node is connected to at least one pin.

The accuracy of the pin assignment model is reported in Table 6, after 150 training epochs.

| Metric | Value (%) |
|---|---|
| Precision | 97.54 |
| Recall | 98.87 |
| F1-score | 98.2 |
| Accuracy | 98.2 |

Table 6: Test set performance metrics of the pin assignment model.

## D   DATASETS STATISTICS

Table 7 presents the main features of the datasets used in this paper. AnalogGenie (no pins) corresponds to the dataset version that is used to train the topology generation model, which is completed by a pin assigment model.

## E   TIME SAMPLING DISTORTION

Following Qin et al. (2025), we apply a *time distortion* function $f$ to the time index $t$ sampled uniformly between 0 and 1. Here we use $f : t \mapsto 1 - (1 - t)^n$, and use the parameter $n$ to control the distortion strength. This procedure allows to control the denoising process by putting an emphasis on critical time steps, e.g. when $t$ approaches 1, where an output circuit can become

| Dataset | OCB | AnalogGenie | AnalogGenie (no pins) |
|---|---|---|---|
| # of graphs | 10,000 | 3,350 | 3,350 |
| % uniqueness | 44.0 | 99.4 | 99.4 |
| Number of node types | 9 | 81 | 28 |
| Avg. # of nodes per sample | 12 | 107 | 38 |
| Avg. # of edges per sample | 15 | 145 | 60 |
| Avg. density | 0.25 | 0.042 | 0.11 |

Table 7: Main dataset statistics.

invalid due to a single edge misplacement. The distortion function is represented in Figure 7 for typical values of $n$ for both training and inference on the AnalogGenie dataset, where a distinct $n$ is used for nodes ($n_v$) and edges ($n_e$).

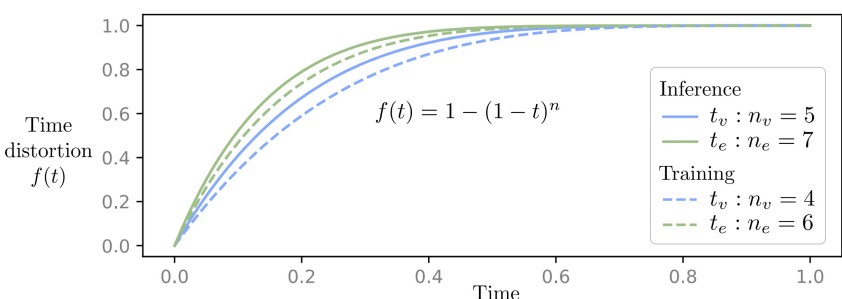

Figure 7: Node ($t_v$) and edge time ($t_e$) sampling schemes on the AnalogGenie dataset.

## F  ILLUSTRATIONS OF DENOISING APPLICATIONS

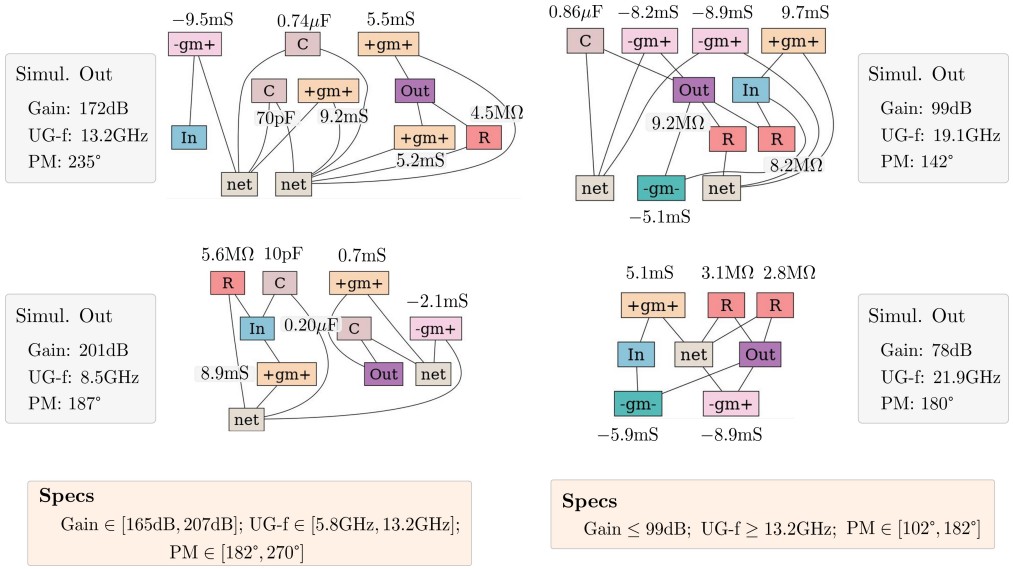

Figure 8: Conditional analog circuit design on OCB.

In this section we illustrate several use cases of CircuitFlow applications, on both OCB and AnalogGenie datasets. Figure 8 pictures two output circuits for two different sets of conditioning specifications on OCB. Examples of transistor-level output topologies can be found in Figures 9 and 10. Then Figures 11, 12 and 13 illustrate the applications of continued denoising, circuit completion, and link prediction, respectively.

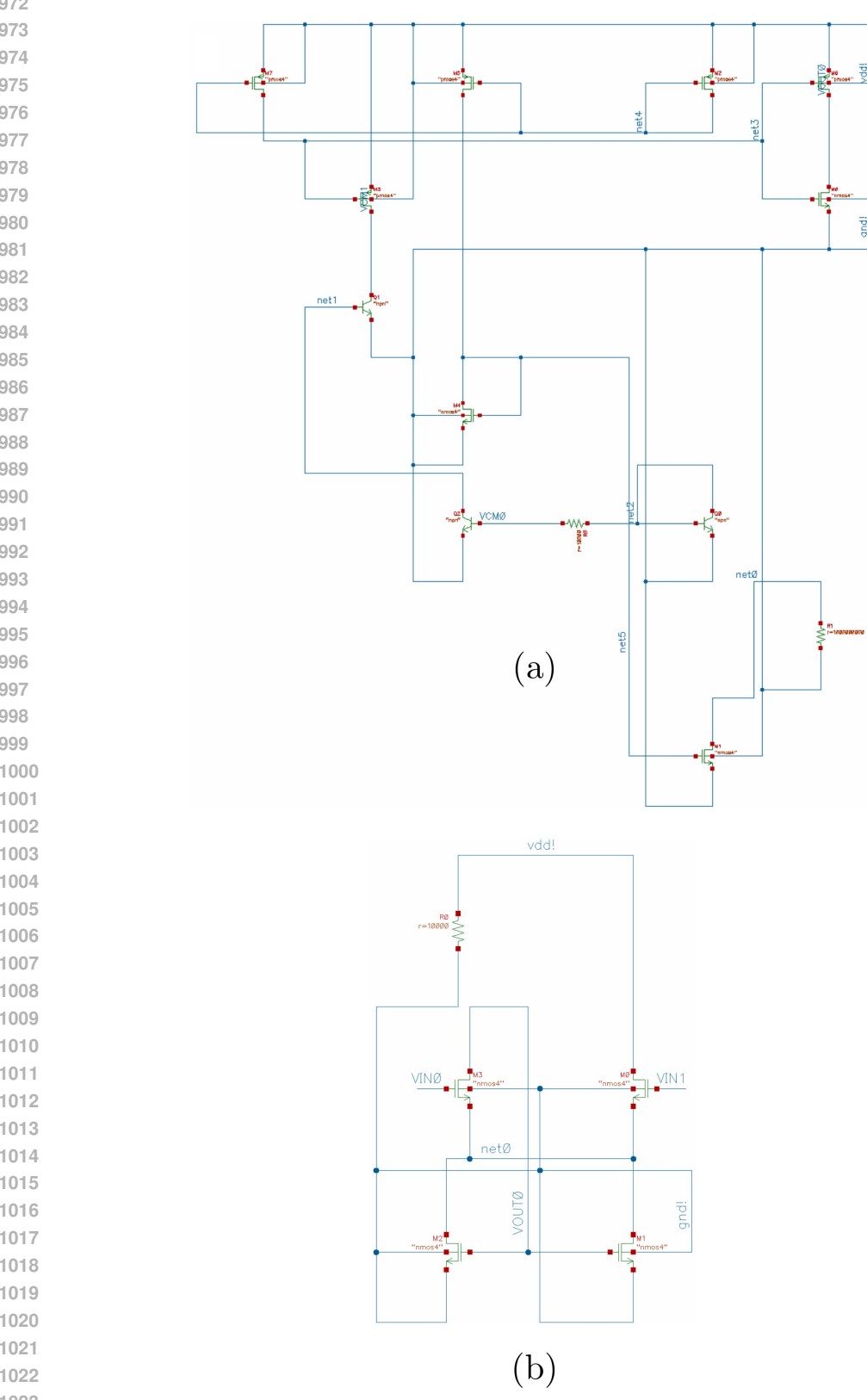

Figure 9: Output topologies generated on the AnalogGenie dataset, showcasing a possible Differential Amplifier (a) and NMOS Logic Gate (b).

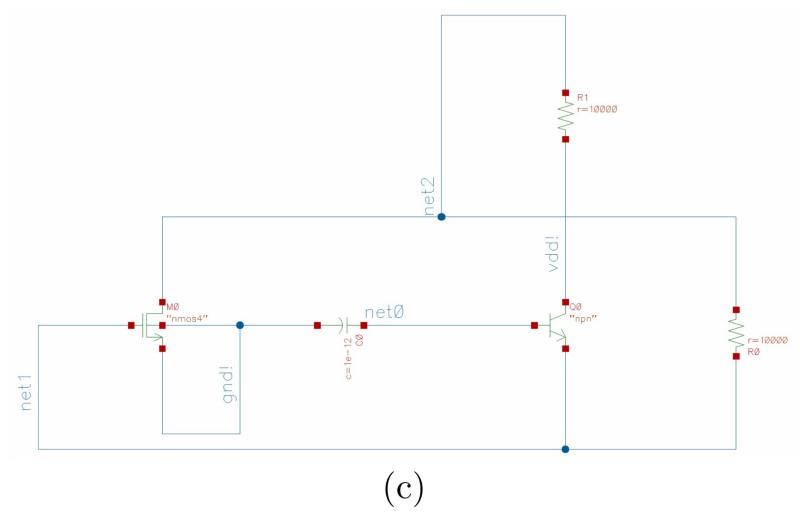

(c)

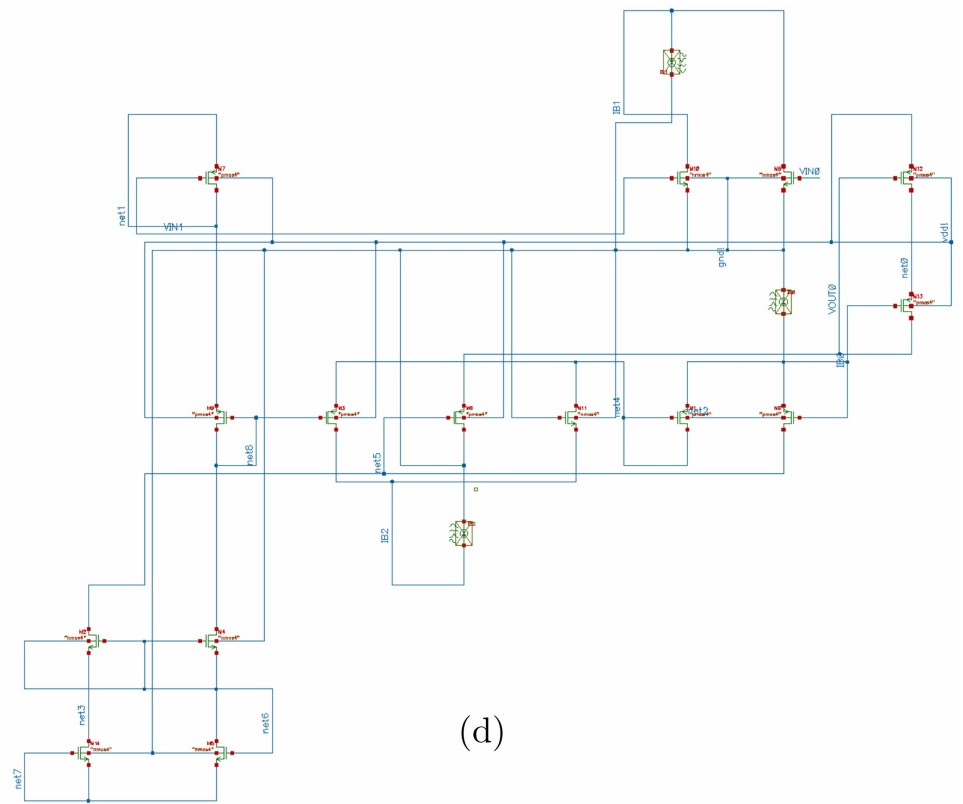

(d)

Figure 10: Additional topologies generated on the AnalogGenie dataset.

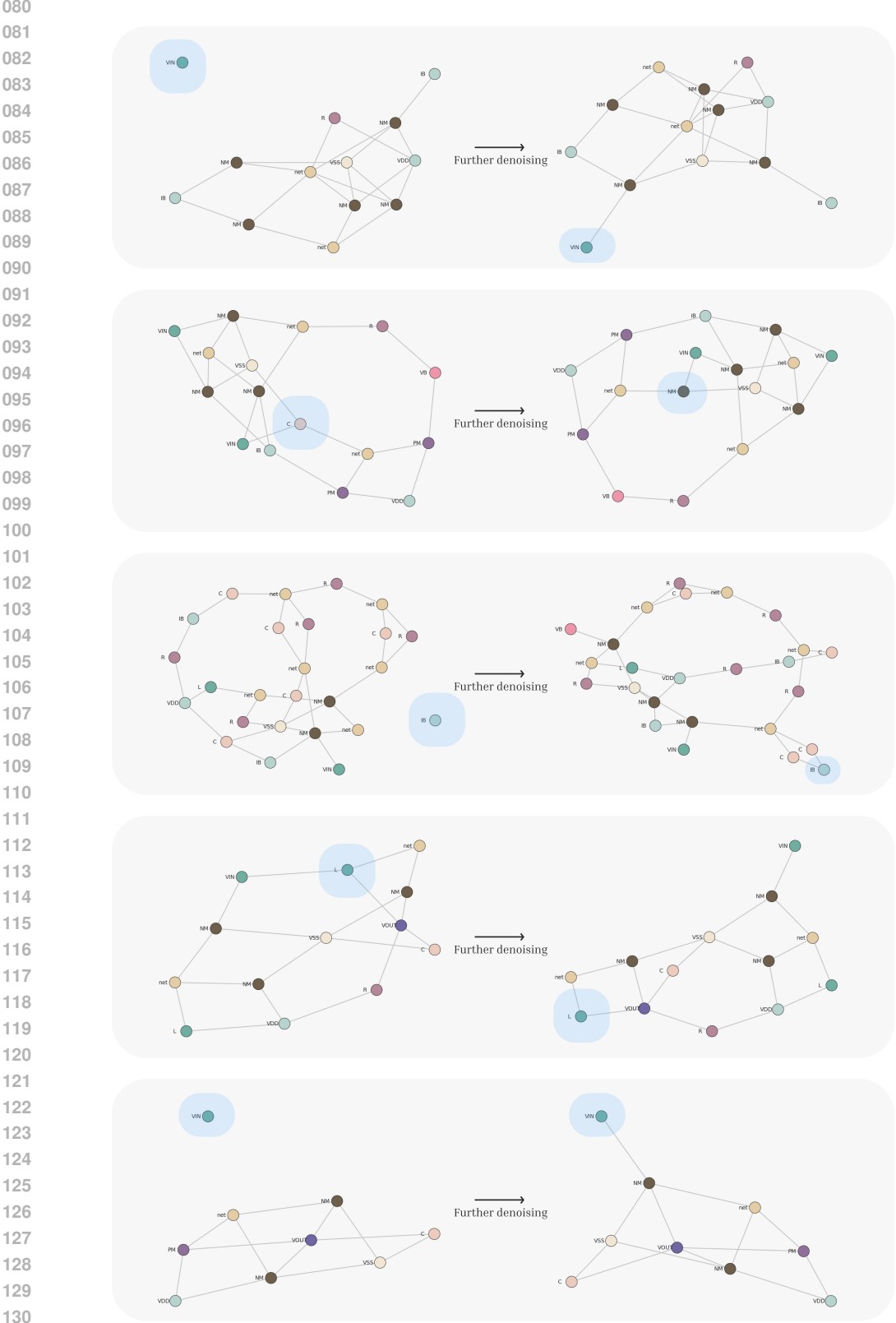

Figure 11: **Continued denoising** can be used to fix topology inconsistencies, as illustrated here on the AnalogGenie dataset.

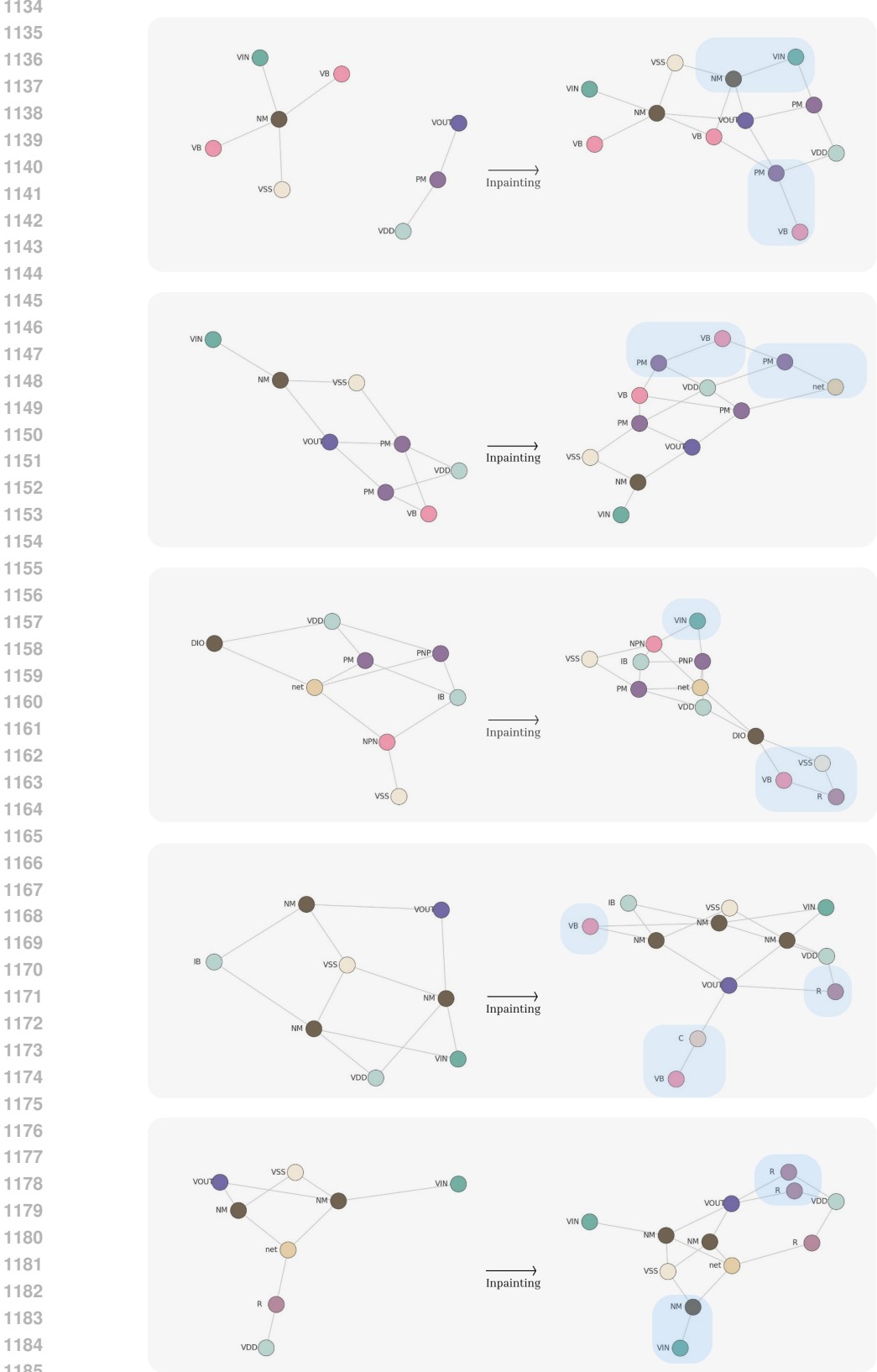

Figure 12: **Circuit completion** examples on the AnalogGenie dataset.

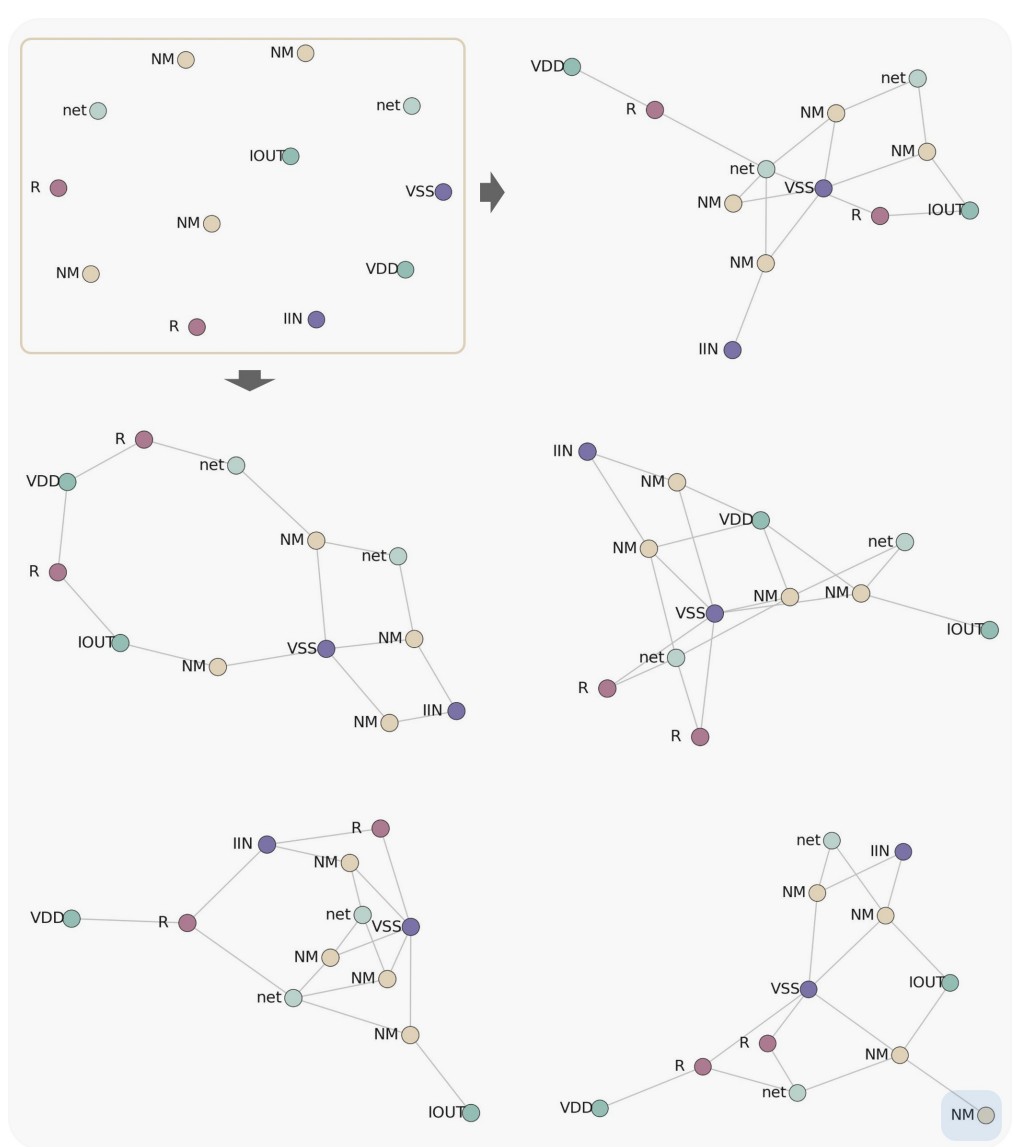

Figure 13: **Link prediction** examples from the same initial *empty graph*, on the AnalogGenie dataset. Generated topologies can include invalid component connections (see bottom right circuit), that could be fixed by additional denoising.

