# OpenReview forum: "Analog Circuit Topology Design and Sizing with Flow Matching Graph Learning"
_ICLR.cc/2026/Conference — Submitted to ICLR 2026_

### Official Review · Reviewer_2PYU · 2025-10-16

**Soundness:** 3
**Presentation:** 3
**Contribution:** 3
**Rating:** 6
**Confidence:** 4

**Summary:**

This work present CircuitFlow, for end-to-end generation of analog circuit topology and device sizing.
This work adopt a multimodal flow matching model and show strong empirical performance on OCB and AnalogGenie dataset.

**Strengths:**

1. One of the first works tackles topology generation and circuit sizing at the same time.
2. Detailed method presentation
3. Using flow-matching on analog circuit front-end design is an underexplored but interesting topic

**Weaknesses:**

1. Lack of strong motivation for unified topology and sizing designs
As the authors point out, there are a lot of dedicated existing works focus on topology or sizing. Why do we need a unified framework? What kind of benefit can it bring?
2. Lack of strong motivation for flow matching
Flow matching is a popular method in image generation. But why do we need flow matching for analog circuit designs, particularly? Does it enable us to generate topology and sizing at the same time compared to the early approach?
3. The current paper describes the above two points as an interesting direction to pursue, but does not show a strong motivation and results on the necessity. It's very hard for me to strongly recommend this paper based on the current material presented.
4. More circuit examples can help the current paper presentation with just graphs

**Questions:**

1. How do you generate different circuits with different sizes (number of devices) with a fixed number of denoising steps?
2. How do you make sure your generated graph is always a single connected component rather than several disconnected components?

---

> ### Author Response · Authors · 2025-11-25
> **Rebuttal by Authors**
>
> We thank the Reviewer for their encouraging comments, and answer below to the questions and remarks that were raised.
>
> **Q1. Lack of strong motivation for unified topology and sizing design**
>
> **R1.** A first answer to this question is the practicality of joint training and inference for topology and device sizes. A second, more mathematically grounded answer can be found in the newly added Table 2 on page 9, where we report the results of two conditional generation methods on the OCB dataset. In particular, we show that control on the generated circuit gain (and phase margin to a lesser extent) can be achieved through *topology generation only*. The practical implication of this result, which indicates that topology is correlated with circuit gain, is that topology generation needs to be done conditionally on the desired specifications, and should not be done with the only aim to maximize the likelihood of (unconditional) output topologies, as CktGNN/AnalogGenie do. From this point of view, jointly generating topology and sizes conditioned on specifications finds a justification.
>
> **Q2. Lack of strong motivation for the use of flow matching**
>
> **R2.** The main motivations for the use of flow matching for analog circuit design are the high sampling quality and the inference-time flexibility, as we mention in the introduction. The sampling quality is illustrated in our experimental results, through the often large improvements obtained over previous works. We give a flavor of sampling flexibility in Section 4.1 of the main text, but also provide extensive illustrations in the appendices (we have added examples on link prediction in Appendix F, to the already existing examples on post-processing/continued denoising and circuit completion). In summary, generating topology and sizes is not unique to flow matching, but it is not clear how previous approaches can accommodate as diverse applications as circuit completion or link prediction.
>
> **Q3. Need for illustrations with circuit examples**
>
> **R3.** We have added examples of output circuits in Figure 4 of the main text and in Figure 8 in appendix.
>
> **Q4. Controlling the number of components**
>
> **R4.** The number of denoising steps is not related to the number of devices, as all nodes / edges are denoised jointly. Following previous works in graph generation (e.g. [1], [2] below) with denoising diffusion / flow matching, we leave node numbers as a parameter that the user is free to set in advance. In our experiments, it is always sampled from the empirical distribution observed from the data (in practice we found very little correlation between node number and circuit validity or conditional generation performances).
>
> [1] Digress: Discrete denoising diffusion for graph generation, Vignac et al., ICLR 23
>
> [2] DeFoG: Discrete Flow Matching for Graph Generation, Qin et al., ICML 25
>
> **Q5. How do you make sure your generated graph is always a single connected component rather than several disconnected components?**
>
> **R5.** This is not enforced in any way but learned from the data. Disconnected graphs are invalid circuits, and therefore the reported metrics already track these inconsistencies.

---

> > ### Comment · Reviewer_2PYU · 2025-11-27
> >
> > I acknowledge the author's thorough rebuttal. My concerns regarding motivations and technical aspects of flow matching methods are addressed. My additional feedback is to include more transistor-level circuit examples in Figure 4 to further strengthen the paper.

---

> ### Author Response · Authors · 2025-11-28
> **Reponse by Authors**
>
> We thank the Reviewer for their feedback, and have added 4 transistor-level generated topologies in Figures 9 & 10 in appendix (they could not fit in Figure 4 given their size). We would gladly clarify any further point or add details that the Reviewer would deem necessary.

---

### Official Review · Reviewer_6r6N · 2025-10-30

**Soundness:** 2
**Presentation:** 1
**Contribution:** 2
**Rating:** 2
**Confidence:** 5

**Summary:**

The paper proposes a flow-based method for both topology generation and device sizing. Extensive experiments show the state-of-the-art performance of the proposed method compared to existing work.

**Strengths:**

The proposed multimodal flow matching framework attempts to unify topology and device generation, which is an interesting direction for analog design automation.

**Weaknesses:**

1. The paper claims (L414–415) to be “the first architecture capable of jointly generating both circuit topologies and device features.”
This is an overclaim, as prior work such as DiffCkt and CktGen [1] has already demonstrated a diffusion-based framework for joint topology-sizing generation.
It seems that the key difference of this work compared to CktGen is that CktGen is a variational autoencoder (VAE) model, while CircuitFlow is based on flow matching.
Section 5.2 does not compare against such state-of-the-art methods, which weakens the experimental credibility and makes it difficult to assess the claimed advantage of the proposed model.

[1] CktGen: Specification-Conditioned Analog Circuit Generation, https://arxiv.org/pdf/2410.00995


2. In Section 5.2, the sizing experiment is underspecified. The training objective maximizes the log-likelihood of device sizes (i.e., matches the dataset distribution), which does not imply performance optimization (gain/pm/ugf) unless the dataset distribution is itself optimal. Evaluation only compares against “data values” and uses a t-SNE plot without defining baseline sampling ranges or reporting quantitative metrics (e.g., validity rate, pass rate with thresholds, Wasserstein/MMD distance, or confidence intervals).
The claim that generated sizes serve as strong BO/GA initialization points is not supported by controlled downstream experiments.

3. Lack of sufficient evidence in the transistor-level experiment. Although Section 5.3 is described as “transistor-level topology generation,” the experiment does not include device sizing, despite the paper’s end-to-end claim.
The reported metrics (V.U.N., validity similarity, uniqueness, novelty, etc.) primarily measure structural diversity and do not demonstrate the functional quality of the generated circuits after sizing or simulation. Thus, the results cannot substantiate the practical utility of the generated designs.

4. Across all experiments, the generation process is not conditioned on circuit performance or design specifications.
For analog front-end design, this significantly limits real-world relevance; generating structurally valid circuits does not imply generating high-quality, spec-satisfying circuits.
The absence of performance-driven evaluation weakens the overall contribution and restricts the framework’s applicability to practical analog design tasks.

5. The comparison with AnalogCoder and AnalogGenie is unfair. How many types of circuits can CircuitFlow generate?

**Questions:**

How does the proposed model handle device sizing or performance constraints during generation?

Can the authors provide post-sizing simulation results (e.g., gain, pm, ugf) to validate circuit quality, especially for the transistor-level generation?

Is it possible to condition the generation process on performance specifications?

How does the method compare with joint topology-sizing models such as CktGen?

What determines the upper limit of the number of devices that CircuitFlow generates?

**Details Of Ethics Concerns:**

N/A.

---

> ### Author Response · Authors · 2025-11-26
> **Rebuttal by Authors**
>
> We thank the Reviewer for their feedback. We believe that it gave us the opportunity to strengthen our work noticeably, especially regarding the control of circuit generation with target performance objectives. Please find below our answers to the different comments and questions.
>
> **Q1. Originality of the task / references and discussions of CktGen and DiffCkt**
>
> **R1.** We agree with the Reviewer, the claim of the originality is incorrect, as joint topology generation and sizing is addressed in the CktGen work. This was made however in good faith: CktGen has been cited only once as of November 2025 and has a low visibility. That said this work indeed shares a very close objective with ours: **we corrected our claim and added references and discussions related to CktGen** (see the Introduction: “*So far, only one study has attempted to tackle these tasks jointly (Hou et al., 2024)*” and Related Work section: “*An exception is CktGen (Hou et al., 2024), which addresses these tasks jointly using a VAE model. Their approach is however limited to operational amplifiers, and the absence of released models precludes formal comparison.*”).
> As stated, the authors of CktGen did not release their model or code, which prevents any comparison or evaluation to compute the missing metrics. It is also *not* possible to report their published results directly: just like CktGNN, with which CktGen shares their dataset and data representation, *ambiguities* in the representation of node connections result in the existence of several SPICE netlists for each produced output (see our Figure 3 and discussion in Section 5.1 on this matter), calling for a necessary processing step before evaluating the different metrics.
>
> DiffCkt, which we mention in the paper, does *not* produce topology and sizes jointly, but uses two separate models for these tasks. Their dataset, which contains only 28 distinct topologies, is however heavily biased towards device sizing. The comparison is further made impractical by the fact that they do not release their code either.
>
>
> **Q2. Sizing experiment underspecified / lack of conditioning on circuit performance**
>
> **R2.** Conditioning circuit generation on expected performances is a major practical use case of circuit design, and we thank the Reviewer for stressing this. This was not originally included in the manuscript because of time constraints, but we conducted additional experiments which proved successful and led us to amend our work substantially.
>
> We explored both **classifier guidance** and **classifier-free guidance** in the context of multimodal flow matching, and obtained satisfactory results in both cases. We summarized our method for performance-conditional circuit generation (including both topology and sizing) in Section 4.3, leaving the details on these classical methods to the appropriate references. Experimental results are reported in Section 5.2 (“Conditional Circuit Design”). These show a **strong effectiveness** of the classifier guidance method, which yields a **joint accuracy on all three objectives** (gain, pm and ugf) **of 28.9%**, out of 64 possible configurations (each quantity is discretized into 4 equally probable quartiles, and we empirically observe that configurations spread over *all* 64 categories).
> **We give examples of generated topologies and sizes in Figure 8 of Appendix F.**
> We believe these new results strengthen our work considerably, enabling the practical application of automatically designing a circuit based on specifications.
>
> Finally, **we updated the provided supplementary files** with the weights of both the unconditional generative model (topology + sizing) and the noisy classifier. A notebook (Classifier_guidance_SUB) allows to reproduce our results step by step, from the generation of circuits conditioned on test specifications to the evaluation of the accuracy (note that the evaluation on the ~900 test samples may take time, around 1hr for us, since the classifier is evaluated at each denoising step on all possible node type transitions).
>
> **Q3. Lack of evidence on the transistor-level experiments**
>
> **R3.** The AnalogGenie benchmark does *not* provide component sizes: we therefore limit our study, at the transistor-level, to topology generation. We agree with the Reviewer that a measure of the practicality of the generated topologies would be beneficial (see however our Addendum to the next question regarding output circuit examples); however in the absence of size information we restrict our evaluations to the executability of the produced SPICE netlists. That said, generating valid topologies on this dataset remains challenging given its high diversity of circuit types and its sample complexity, albeit being relatively small-scale.

---

> ### Author Response · Authors · 2025-11-26
> **Rebuttal by Authors - continued**
>
> **Q4. Unfair comparison with AnalogCoder / AnalogGenie, number of circuit types?**
>
> **R4.** The only difference between our experimental protocol and that of AnalogGenie is that we do not report the number of output circuit types. However its evaluation is questionable, and the authors of AnalogGenie do not provide the code for the evaluation of their metrics. As far as we know, it might as well be qualitatively evaluated by a visual inspection of output results, which raises concerns on the robustness of the evaluation protocol. It is however possible to measure quantities that serve as a proxy for the diversity of generated graphs. One is the Jensen-Shannon divergence between the data and output node type distributions, that we calculated on our results and on circuits produced by AnalogGenie.
> The results were added to Table 3 in the paper. The JS we obtain for CircuitFlow ($1.8$ and $2.6e^{-3}$) are an order of magnitude lower than that of AnalogGenie outputs ($14.5e^{-3}$): it shows that our flow matching approach has learned much more closely to generate circuit components based on their relative appearance in the dataset, which suggests that the training dataset is itself more faithfully reconstructed.
>
> *__ADDENDUM November 28th__* - We have added Cadence schematics of 4 transistor-level analog topologies produced by our model in Figures 9 and 10 (Appendix F), that we considered illustrative of the diversity of the generated circuits. Although assessing their practicality in the absence of sizes is challenging, they can be reminiscent of well-known analog IC building blocks (here for instance logic gate / differential amplifier).
>
> **Q5: Post-sizing simulation illustrations**
>
> **R5.** In Figure 8 (Appendix F), we included 4 output circuit examples, conditioned on two sets of specifications, along with the corresponding simulation outputs.
>
> **Q6. What determines the upper limit of the number of devices that CircuitFlow generates?**
>
> **R6:** There is no theoretical upper limit on the number of devices that CircuitFlow can generate, which is a parameter that the user is free to set in advance (following previous literature on graph generation). That said, circuits with a number of nodes that deviates too much from the data distribution are likely to be of lower quality simply because the network has not learned to model them properly.

---

### Official Review · Reviewer_Pf41 · 2025-11-01

**Soundness:** 3
**Presentation:** 2
**Contribution:** 3
**Rating:** 6
**Confidence:** 4

**Summary:**

This paper presents CircuitFlow, a multimodal flow-matching model for joint analog circuit topology generation and device sizing. It introduces dimension- and modality-wise time sampling to handle heterogeneous variables and uses a graph transformer backbone. With a unified graph representation and a separate pin-prediction module, the method reports state-of-the-art results on OCB and AnalogGenie, and performance can be further improved via a simple continued-denoising post-process.

**Strengths:**

+ This paper proposes a unified multimodal flow-matching formulation for analog circuit topology and sizing, which is novel and well-motivated.
+ The CircuitFlow framework supports flexible inference tasks (e.g., generating new topologies or editing partial structures), which is practically useful.
+ Experiments show state-of-the-art results on OCB and AnalogGenie, with additional gains from a lightweight post-processing step.

**Weaknesses:**

- The paper appears to overclaim on sizing. In Table 2, comparisons are limited to data vs. CircuitFlow vs. random, where CircuitFlow’s gain and phase margin are below data, which cannot support the claim that the model has “indirectly learned to maximize all three properties.” And it does not compare the difference between the predicted sizes and the real optimized sizes.
- Although the method section emphasizes the dimension- and modality-dependent time sampling scheme design choice, there is no ablation study demonstrating its specific contribution.
- Some important experiments are missing. The influence of different circuit graph representations (with or without preprocessing). For the two-stage pin assignment, it is good to have an ablation study for unified v.s. separate processing.
- The paper uses substantial space to restate the known formulas and preliminaries, but provides insufficient implementation detail for the proposed method itself. For example, a concrete algorithm for the dimension- and modality-  during training and inference would be helpful.

**Questions:**

For the AnalogGenie (no pins), there are only ~38 nodes and ~60 edges per sample, but you use 1000 denoising Euler steps. Why use so many steps for inference? What is the impact of fewer steps on quality and runtime? Please report inference latency for different inference step settings and compare with the baseline.

---

> ### Author Response · Authors · 2025-11-25
> **Rebuttal by Authors**
>
> We thank the Reviewer for their insightful and positive comments. Below we answer to the different remarks and questions.
>
> **Q1. Limited evidence for sizing**
>
> **R1.** We used the time frame offered by the rebuttal period to *rework this section in depth*. Two conditioning mechanisms from the denoising score matching literature (namely classifier guidance and classifier-free guidance) have been adapted to unlock conditional generation for our framework. The method is described in a newly appended paragraph of the Method section (Section 4.3 “Conditional Generation”), and results are provided in the “Conditional Circuit Design” paragraph of Section 5.2. These show that **fine-grained control can be achieved** on Gain, Phase margin and Unit-gain frequency through both topology generation and sizing. Examples of output circuits and sizes for different sets of conditions have been added in Figure 8 (Appendix F).
>
> **Q2. Ablation on the effects of modality-/dimension-wise time sampling**
>
> **R2.** We have added an ablation study on the various time sampling schemes at training and inference (Table 4, section 5.4.). These show that allowing dimension-wise $t$ sampling during training improves the validity for applications where one *needs* to consider different time indices for different dimensions. This is the case of circuit completion, where the denoising time index of a fraction of nodes and edges, corresponding to the circuit that must be completed, is fixed to 1.
>
> **Q3. Missing experiments on the effects of preprocessing and of the hierarchical approach for pin assignment**
>
> **R3.** Failing to preprocess the data, especially in our approach which works on undirected graphs, leads to topology ambiguities: without representing net nodes, it is not possible to tell how a circuit component is connected to its neighbors (e.g. distinguishing input/output neighboring components). The preprocessing is therefore necessary. In Figure 3 we illustrate how topology ambiguities (both due to the absence of net nodes and to the DAG representation) can lead to train on and generate invalid topologies.
>
> In this example generated by CktGNN, the capacitor is in parallel with a feedback op-amp, and both belong to a feedback path which connects the *output of the first feedforward op-amp* with $V_{out}$. From the DAG representation of CktGNN, one would however wrongly interpret the same capacitor as connecting the *output of the second feedforward op-amp* with $V_{out}$ (and thus consider the circuit as valid). An undirected graph representation with explicit net nodes fully disambiguates such problematic cases.
>
> Regarding the two-stage pin assignment model, our experiments suggest that directly predicting all pin connections along with other components within a single model leads to almost zero validity. This is due to the large size of AnalogGenie graphs, causing even a single wrong prediction in pin assignment to make the graph invalid.
>
> **Q4. Algorithm missing**
>
> **R4.** Training and sampling algorithms have been added in Section 4.1. Dimension-wise time sampling is mentioned implicitly in the dimension of the sampled time index ($2D_v + D_e$). However, it does not affect the training / sampling procedures further.
>
> **Q5. Impact of Euler steps number on performances and inference runtime**
>
> **R5.** An ablation table on the impact of the number of denoising Euler steps on circuit quality and inference latency has been added in Section 5.4. Further hyperparameters tuning (which we detail in Appendix E) let us improve on previous results using as few as 100 Euler steps, and our study shows that those are rather robust to the precise number of steps. In fact, using 50 denoising steps already improves VUN by 10 points over AnalogGenie, while being 26 times faster.

---

### Official Review · Reviewer_YrVy · 2025-11-01

**Soundness:** 2
**Presentation:** 2
**Contribution:** 2
**Rating:** 4
**Confidence:** 4

**Summary:**

The paper proposes a generative model for analog circuits using multimodal flow matching. It jointly generates (1) circuit topology (discrete graph: devices, nets, connections) and (2) device sizing parameters (continuous values). The key claimed contribution is a per-dimension time schedule: each node, edge, and sizing parameter is assigned its own noise time, which the authors say enables conditional tasks like partial completion and repair. The model is evaluated on two circuit benchmarks and reports high validity / uniqueness / novelty / simulability. The problem is important. Automatically generating spec-worthy analog circuits (both topology and sizing) is a major goal in analog EDA.

**Strengths:**

(1) Tackles a high-impact task (automatic analog circuit synthesis).

(2) Attempts joint topology + sizing generation with one model, which is practically valuable.

(3) Claims ability to “edit/repair” subcircuits or size a fixed topology by selectively denoising only parts of the graph.

(4) Shows promising validity/simulability numbers.

**Weaknesses:**

(1) The model is largely an application of known multimodal flow matching (discrete jump process for structure + continuous rectified flow for sizing). The only new mechanism claimed is the per-dimension time scheduling. This looks like an incremental extension of the standard factorized noise assumption, not a fundamentally new objective, and it is currently oversold.

(2) **No ablation for the claimed contribution.**
There is no experiment showing that per-dimension time scheduling actually matters. We need a direct comparison against:

(A) single global time for all dimensions,

(B) per-modality time (one for topology, one for sizing),

(C) the proposed per-dimension time (one per node/edge/parameter).


(3) **Architecture underspecified:**
The paper does not clearly explain how the model enforces circuit legality:

(3.1) how illegal device types / pin assignments are prevented,

(3.2) how symmetric structures (current mirrors, differential pairs) are handled,

(3.3) how the second-stage pin assignment module works at inference.

These constraints are central to analog design and must be described mathematically to make the work reproducible.

(4) **No design-yield / success rate.**
Analog design is a constrained optimization problem, not just “generate something simulatable.” The paper does not report the core metric an analog designer cares about: if you sample N times (e.g. 5–10), what fraction of generated circuits are (i) topologically valid, (ii) fully sized, and (iii) satisfy all required specs (gain, bandwidth/UGBW, phase margin, power) with no post hoc tuning?
Without this “yield,” it’s unclear if the method is actually designing usable circuits or just producing plausible sketches.

(5) **Baselines are weak.**
Apart from CktGNN, most baselines are generic generative models or author-modified references not built for analog circuit synthesis. This makes the reported gains less convincing. The paper does not compare against realistic circuit design pipelines such as:

topology generation + RL/BO/SPICE sizing loops,

retrieval + constraint-guided GNN repair,

LLM-generated SPICE + filtering.
Claiming state of the art without these comparisons is premature. (refer the following work for baselines or related works):

[Related Works]:
- Graph of circuits with GNNs for exploring optimal design space
- Learning to Design Analog Circuits to Meet Threshold Specifications
- GANA: Graph Convolutional Network Based Automated Netlist Annotation for Analog Circuits


(6) **Related work is incomplete.**
The paper under-cites prior work that already treats circuits as graphs and uses GNNs / RL / optimization to generate or refine analog topologies and size them to meet specs. These should be cited and, where possible, used as baselines.

(7) **Metrics are poorly defined**:
“Validity,” “uniqueness,” “novelty,” and “max node count” are reported but not rigorously defined or tied to actual analog design criteria. It’s unclear:
whether “validity” means “SPICE runs,” or “meets spec,”
how “uniqueness” handles isomorphic netlists,
what level of change counts as “novel,”
what “max node count” means in terms of real design complexity.
As written, these look like generic graph-generation metrics, not design quality metrics.

**Questions:**

Refer to weaknesses

---

> ### Author Response · Authors · 2025-11-27
> **Rebuttal by Authors**
>
> We thank the Reviewer for their review, and answer below to the points that were raised.
>
> **Q1. Interrogation on the significance of the contribution**
>
> **R1.** This is indeed an *application* paper, of an increasingly popular but still very recent methodology (at least concerning flow matching on discrete state space and for graph generation), to a domain where we believe it can have a high impact. Indeed, we show substantial improvement over previous state of the art on all tested benchmarks and experimental configurations, and this is our first and foremost contribution.
>
> The methodological novelty we propose, which is our second contribution, allows a whole range of practical applications for analog circuit design that we detail and illustrate extensively (see Section 4.1 and Figure 1, but also Figures 11, 12 and 13 in Appendix F).
>
> **Q2: Ablation on the effectiveness of per-dimension time sampling**
>
> **R2.**  Thank you for this suggestion: we conducted an ablation on different time sampling schemes, both during training and inference. It shows that for an application like circuit completion that *requires* dimension-wise $t$ sampling, the model trained with the same time sampling scheme yields the best results (+3 pts of validity). Finally, training by sampling the same time index for the whole graph consistently yields poorer results (se Table 4 in Section 5.4 - *Ablation Studies*).
>
> **Q3. Underspecified architecture**
>
> **R3.** Legality is not enforced but learned throughout our experimental configurations. In fact, no post-processing or inductive bias is ever applied or implemented apart from what is described in the paper: a circuit representation that disambiguates node connections, and a post-processing limited to additional denoising steps.
> We have added details to the functioning of the second-stage, pin assignment module in Appendix D, highlighting the combinatorial optimization nature of the assignment problem. Regarding the reproducibility, all the models will be released open-source, making the work fully reproducible.
>
> **Q4. No design-yield / success rate.**
>
> **R4.** Thank you for your remark. We already report the success rate regarding topology validity (along with novelty and uniqueness), but were missing the important topic of adequacy with pre-set circuit performances. *We have added a substantial section on conditional generation* (see Section 5.2, paragraph “Conditional Circuit Design”), presenting precisely the results of accuracy measures on the obtained gain, phase margin and unit-gain frequency, compared to specifications (our methodology is described in Section 4.3). To do so, all three quantities were quantized into four equally probable categorical bins. Our best results achieve *28.9% joint accuracy* over test set specifications, but also $\sim 73$% accuracy on phase margin,  $\sim 66$% on gain and  $61$% on ugf (all of this is done by generating a *single* output per specification).
>
> **Q5. Baselines are weak.**
>
> **R5.** We respectfully disagree. CktGNN and AnalogGenie are **current state-of-the-art models** on their respective datasets, which are themselves standard benchmarks for the topology generation task, addressing various aspects of analog circuit design and circuit representations. AnalogGenie in particular (ICLR’25 spotlight) successfully tackles the very challenging task of modelling 11 different analog circuit *categories* (corresponding to ~3300 unique topologies, some being highly complex) with a *single model* and is definitely **not a weak baseline**. The additional baseline methods mentioned in the experiments are the **exact same** that were previously reported in the two aforementioned works, making our experimental protocol fully in line with the literature.
>
> **Q6. Related work is incomplete**
>
> **R6.**
> We thank the Reviewer for pointing out the work of Shahane et al., which we now mention in the introduction with other GNN methods for sizing.
>
> GANA, on the other hand, is a circuit identification framework and is largely *irrelevant* in the context of our task.
>
> Finally, we *already cite* the work of Krylov et al, which is restricted to device sizing on a limited set of 7 given circuit topologies, and is thus inapplicable in our context.
>
> Although we tried to be as exhaustive as we could, we would be glad if the Reviewer could indicate any additional reference that we would have missed.
>
> Finally, we want to stress that none of the three mentioned references generate or even refine analog topologies, making them inapplicable as baselines.

---

> ### Author Response · Authors · 2025-11-27
> **Rebuttal by Authors - continued**
>
> **Q7. Metrics are poorly defined**
>
> **R7.** All those metrics are established metrics in the literature of circuit (and graph) generation. Validity metrics *are defined* in the paper: “*percentage of connected graphs with input and output nodes*” for **Valid graph**, “*circuit with a main path composed only of single-stage op-amps*” for **Valid circuits**, and “*proportion of circuits simulatable with SPICE with default parameters*” for **Valid sim**. We expressly mention the origin of these metrics in the text (Section 5.2), **CktGNN** for the first two and **AnalogGenie** for **Valid sim**. **Novelty** (used in both previous works), **Uniqueness** and **VUN** are classical metrics in the graph generation literature, and although not being explicitly related to circuits, give precious information on the quality of the generated topologies.

---

### Meta-Review · Area_Chair_myZp · 2026-01-07

**Summary:**

This paper proposes CircuitFlow, a multimodal flow matching generative model for joint analog circuit topology generation and device sizing, with a graph transformer backbone and a two-stage pin assignment module. The claimed technical contribution includes modality-/dimension-wise time sampling enabling flexible inference tasks such as completion/repair, and the paper reports strong results on OCBand AnalogGenie. Several reviewers agree the direction is promising and the empirical gains on benchmark generation metrics are substantial.

However, the discussion also reveals a set of core scientific and evaluation risks that remain insufficiently resolved for an ICLR acceptance—primarily around (i) overclaiming and incomplete positioning vs prior joint topology+sizing work, (ii) the practical meaning of reported metrics for analog design (design yield/spec satisfaction), and (iii) reproducibility/clarity of the method and evaluation protocols. While the rebuttal adds missing ablations and introduces a conditional generation experiment, those additions still leave the overall impact and real-world relevance ambiguous, and one high-confidence reviewer remains strongly negative. Given the remaining disagreement and the fact that the submission’s strongest evidence is still in benchmark-style graph-generation metrics rather than a robust spec-driven design-yield evaluation, I recommend rejection.

**Reviewer Concerns:**

**Addressed by rebuttal / revisions**
- Ablation for time-sampling scheme . Multiple reviewers requested ablations comparing global-time vs per-modality vs per-dimension time schedules. The authors report adding an ablation and claim improvements in completion-style settings when training uses the same dimension-wise time scheme as inference.
- Conditional generation toward performance specifications (gain/PM/UGF). A key criticism was the lack of performance-conditioned design. The authors added a “Conditional Circuit Design” section with classifier(-free) guidance and report quartile-binned accuracy (including a reported 28.9% joint accuracy over 64 bins) plus example circuits and post-sizing simulation illustrations in the appendix.
- Runtime concerns for AnalogGenie inference steps. Authors add an ablation showing fewer Euler steps (e.g., 50–100) can maintain quality and reduce latency, with claims such as being substantially faster than AnalogGenie while improving VUN.
- Acknowledgement and correction of an overclaim about being “first” to jointly generate topology + sizing. Authors explicitly retract the strongest novelty claim and add discussion of CktGen, noting lack of released model/code prevents direct comparison.

**Still outstanding / decisive concerns**
- Spec satisfaction / design yield remains weakly demonstrated.
Even with the new conditional-generation experiment, the evaluation remains coarse (quartile bins) and reports “accuracy” rather than the designer-relevant yield: e.g., fraction of generated circuits meeting numeric spec thresholds (gain/UGBW/PM/power) with realistic tolerances, under limited sampling budgets, and without extensive post-hoc tuning. A 28.9% joint bin accuracy (single sample per spec) is hard to interpret as an actionable design yield, and it is unclear how it compares to baseline design strategies (e.g., optimization-based sizing loops, BO/GA initialization baselines, or guided search methods) under comparable compute budgets.
- Baseline and positioning gaps remain, especially for joint topology+sizing.
The lack of direct comparison to CktGen/DiffCkt is partially excused by missing code, but that still leaves a central uncertainty: whether the reported gains are due to representation differences, dataset preprocessing, or the proposed flow-matching formulation. For an ICLR paper claiming SOTA on end-to-end design, the absence of a strong joint baseline (or a careful, reproducible proxy) significantly weakens the credibility of “state-of-the-art” claims for the joint task.
- Method clarity / reproducibility still relies heavily on “we will release code.”
Reviewer YrVy and others questioned underspecified aspects: how legality is ensured, how pin assignment works, and what algorithmic details are needed to reproduce training/inference (especially for time sampling). The rebuttal adds some details, but key parts still read as: legality is “learned,” and reproducibility will come from code release. For evaluation at ICLR, the paper should stand on its own with sufficiently precise descriptions.
- AnalogGenie experiment is topology-only; practical utility remains unclear.
For AnalogGenie, there is no sizing information, and evaluation is mostly executability / VUN-style metrics. Without post-sizing simulation or functionality assessment, the practical relevance of transistor-level generated topologies is still uncertain.
- Persistent concerns from high-confidence negative reviewer(s).
Reviewer 6r6N (rating 2, confidence 5) remains the most concerning: they challenge the end-to-end claim, fairness/coverage vs analog design goals, and demand post-sizing simulation and performance conditioning. The rebuttal addresses part of this (conditional generation on OCB, examples), but the reviewer did not update their score and many of their concerns—especially about design relevance and missing comparisons—remain open at decision time.

**Reviewer Scores:**

- YrVy (4 → 6): The new ablation and conditional-generation section would likely move this reviewer upward slightly, though they might still remain borderline given concerns about design-yield and baseline strength.
- Pf41 (6 → 6 ): This reviewer’s major requests were addressed. Likely stays positive, possibly improving.
- 2PYU (6 → 6): Reviewer explicitly states key concerns were addressed and remains positive.
- 6r6N (2 → 4 at most): Even if discussion occurred, the core critique is only partially answered. I would expect at best a small move upward, remaining below threshold.

Given this expected spread, the panel would likely remain split with at least one highly confident negative reviewer and multiple reviewers still expressing “would not mind if rejected.” In such a scenario, the burden is on the paper to present unambiguous, design-relevant evidence, which is not fully met here.

---

### Decision · Program_Chairs · 2026-01-26

Reject